# Haystack Engineering: Context Engineering for Heterogeneous and Agentic Long-Context Evaluation

## Abstract

Modern long-context large language models (LLMs) perform well on synthetic "needle-in-a-haystack" (NIAH) benchmarks, but such tests overlook how noisy contexts arise from biased retrieval and agentic workflows. We argue that **haystack engineering** is necessary to construct noisy long contexts that faithfully capture key real-world factors—distraction from heterogeneous biased retrievers and cascading errors in agentic workflows—to test models' long-context robustness. We instantiate it through **HaystackCraft**, a new NIAH benchmark built on the full English Wikipedia hyperlink network with multi-hop questions. Haystack-Craft evaluates how heterogeneous retrieval strategies (e.g., sparse, dense, hybrid, and graph-based) affect distractor composition, haystack ordering, and downstream LLM performance. HaystackCraft further extends NIAH to dynamic, LLM-dependent settings that simulate agentic operations, where models refine queries, reflect on their past reasonings, and decide when to stop. Experiments with 15 long-context models show that (1) while stronger dense retrievers can introduce more challenging distractors, graph-based reranking simultaneously improves retrieval effectiveness and mitigates more harmful distractors; (2) in agentic tests, even advanced models like Gemini 2.5 Pro and GPT-5 suffer cascading failures from self-generated distractors or struggle to perform early stops. These results highlight persistent challenges in agentic long-context reasoning and establish HaystackCraft as a valuable testbed for future progress.

## 1 Introduction

Effective context engineering (Mei et al., 2025)—optimizing information for LLMs' contexts—and robust long-context reasoning are essential for large language models (LLMs) to enable sophisticated agents and handle complex, information-intensive tasks. Recent algorithmic and engineering innovations have significantly expanded LLMs' context windows and enhanced their long-context reasoning capabilities (Su et al., 2024; Peng et al., 2024; Dao et al., 2022; Dao, 2024; Liu et al., 2024a; Xiao et al., 2024; Yuan et al., 2025; Kwon et al., 2023). Consequently, modern LLMs can process extended contexts and often achieve near-perfect recall on synthetic "**needle-in-a-haystack (NIAH)**" benchmarks (Yen et al., 2025). These benchmarks test whether a model can retrieve and reason over relevant information *needle* buried in a large *haystack* context that also contains many distractors. Yet such synthetic setups neglect a fundamental question: how are noisy long contexts constructed in real-world applications?

To engineer long contexts in practice, retrieval-augmented generation (RAG) (Lewis et al., 2020) is one of the most widely adopted strategies, where external retrievers rank candidate context documents with respect to queries. However, retrieval systems are imperfect and inherently biased, introducing retriever-dependent ranked distractors. To be specific, sparse retrievers such as BM25 (Robertson et al., 1994; Robertson & Zaragoza, 2009) often populate haystacks with lexically similar but semantically irrelevant documents, while dense retrievers (Karpukhin et al., 2020) surface semantically close but potentially factually incorrect "near misses". Because no single retriever is universally optimal (Thakur et al., 2021), it is crucial to study how heterogeneous retrieval strategies shape the context and consequently affect NIAH performance.

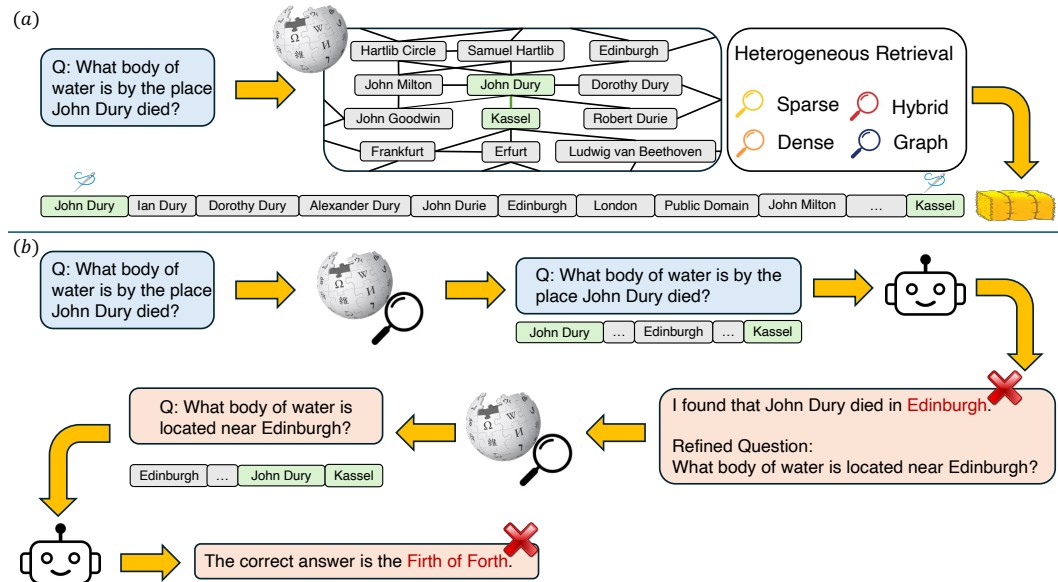

Figure 1: Overview of the core challenges that HaystackCraft addresses. (a) **Retrieval-Dependent Haystacks.** The composition and ordering of the noisy long context ("haystack") are shaped by the retrieval strategy (e.g., sparse, dense, hybrid, and graph-based). (b) **Agentic Error Propagation.** In dynamic agentic workflows, early errors—such as misidentifying John Dury's death place—can propagate through query refinements. This leads to cascading failures where the agent deviates from the original query's intent and inflates distractor rankings.

Beyond heterogeneous retrieval biases, interdependencies between needles and distractors introduce another layer of complexity. Many queries are multi-hop, which requires connecting multiple scattered, logically related needle pieces (Yang et al., 2018). In networked corpora such as social or hyperlink networks, these pieces are explicitly connected to each other and to potential distractors, yielding the challenge of identifying a smaller "needle subgraph" within a larger "haystack graph". Graph-based retrieval methods are central to modern information retrieval and search engines (Page et al., 1999). Collectively, these factors create a realistic and nuanced interplay that has been largely overlooked in prior NIAH studies (Fig. 1 (a)).

Furthermore, advances in reasoning LLMs (OpenAI, 2024; DeepSeek-AI et al., 2025) enable increasingly LLM-driven agentic context engineering (Yao et al., 2023) for challenging tasks like DeepResearch (Google, 2025). Rather than passively digest provided contexts and directly jump to the conclusion, LLMs can perform multi-round research (Trivedi et al., 2023; Jiang et al., 2023), actively refining the query to improve the retrieval quality (Wang et al., 2023; Ma et al., 2023) and reflecting on their own past reasoning (Shinn et al., 2023; Asai et al., 2024). In such dynamic systems with adaptive retrieval and iterative reasoning, LLMs become a distractor source themselves. Early-stage errors—such as noisy retrievals or hallucinated facts—can propagate and compound, leading to cascading failures over iterations or gradual deviation from the original query intent (Fig. 1 (b)). As a result, static, single-round, and LLM-independent NIAH evaluations are insufficient, motivating dynamic, multi-round, LLM-dependent test environments.

In order to mitigate the simulation-to-reality gap for LLMs' long-context utility, we argue that **haystack engineering** is necessary. Just as context engineering seeks to provide optimal LLM contexts, haystack engineering addresses the challenge of constructing realistic noisy long contexts. While context engineering emphasizes best-case conditions, haystack engineering emphasizes faithful haystack constructions shaped by heterogeneous retrieval strategies and cascading agentic errors.

We instantiate this concept through **HaystackCraft**, a new NIAH benchmark built on the full English Wikipedia hyperlink network and multi-hop questions. HaystackCraft systematically examines how retriever choice shapes distractor composition, haystack ordering, and the resulting LLM performance. It evaluates widely adopted retrieval strategies, including sparse, dense, hybrid, and graph-based methods. To contextualize the evaluation in agentic context engineering, we introduce novel LLM-dependent, dynamic NIAH tests. Featuring crucial agentic operations like query

refinement and summarization, HaystackCraft challenges models in two dynamic long-context settings: (1) an enforced multi-round scenario to measure robustness against cascading errors and (2) a variable-round scenario to examine if models can proactively escape cascading errors by early stop.

We perform extensive studies covering 15 long-context LLMs, featuring general-purpose, reasoning, open-source, and commercial models. First, we find that retrieval strategy strongly impacts haystack difficulty. While dense retrievers introduce harder distractors than sparse ones, graph-based reranking with Personalized PageRank (PPR) simultaneously improves retrieval effectiveness and mitigates more harmful distractors, improving NIAH performance by up to 44%. Second, our dynamic NIAH tests reveal that current models are surprisingly brittle in agentic workflows. Even advanced models like Gemini 2.5 Pro and GPT-5 suffer from cascading self-distraction when multiple reasoning rounds are enforced. Crucially, models tend to be more robust to single-round noisy long contexts ("width") than to noisy reasoning iterations ("depth"). Even when models are allowed for an early stop, most models fail to terminate the process appropriately. Overall, our evaluations suggest that long-context challenges in realistic, agentic context engineering are far from solved—and that HaystackCraft provides a valuable testbed for measuring progress on these issues.

## 2 RELATED WORK

**Long-Context Benchmarks.** The original NIAH test inserts a single needle sentence into increasingly large haystacks (Kamradt). LV-Eval (Yuan et al., 2024), RULER (Hsieh et al., 2024), and BABILong (Kuratov et al., 2024) extend the test in terms of question types and corpus sources. However, all these attempts construct query-independent distractors, rather than retriever-dependent contexts, as in practical applications like RAG. HELMET (Yen et al., 2025) takes a step toward realism by using a dense retriever for distractor construction, but it does not capture retriever heterogeneity, network-structured corpora, or the influence of retriever-ranked haystack ordering. Beyond NIAH, other benchmarks assess long-context reasoning in downstream tasks and domain-specific applications (Shaham et al., 2022; Dong et al., 2023; Shaham et al., 2023; An et al., 2024; Bai et al., 2024a;b; Wang et al., 2024; Zhang et al., 2024; Wang et al., 2025). However, these benchmarks lack the flexibility that NIAH provides in context size and distractor composition. Furthermore, unlike HaystackCraft, all existing long-context benchmarks employ static, LLM-independent contexts. This approach is insufficient for evaluating LLMs in dynamic, multi-round agentic systems.

**Agentic Search.** Benchmarks such as BrowseComp (Wei et al., 2025) and Search Arena (Miroyan et al., 2025) evaluate LLMs performing multi-step, agentic web search. As in RAG settings, retrieved information can be irrelevant, distracting, or non-factual (Yoran et al., 2024; Jin et al., 2025), which can degrade agent performance. Moreover, operating over the open web also introduces risks of search-time data contamination, where answers to benchmark questions may surface directly in results (Han et al., 2025). HaystackCraft presents a complimentary effort by studying the intersection of agentic search and long-context reasoning in a controlled environment with a fixed corpus.

## 3 HAYSTACKCRAFT

Context engineering aims to select, structure, and optimize an LLM's input context to maximize its reasoning effectiveness (Lewis et al., 2020; Yao et al., 2023; Mei et al., 2025). Its practice shapes LLMs' long-context challenges. We introduce the complementary concept of **haystack engineering**: the principled construction of realistic noisy long contexts that faithfully model the complexities and failure modes of real-world context engineering pipelines. While context engineering seeks to improve performance, haystack engineering aims to create challenging test conditions to measure model robustness, shaped by factors like heterogeneous retrieval strategies and cascading errors in agentic workflows. We present **HaystackCraft**, a benchmark that instantiates this principle.

In this section, we first formalize the NIAH problem arising from RAG, which highlights the central role of the retrieval strategy and motivates studying representative, heterogeneous retrievers. We then introduce, to our knowledge, the first dynamic, LLM-dependent NIAH challenge, designed to characterize long-context challenges in agentic context engineering. Finally, we describe how HaystackCraft is grounded in the full English Wikipedia hyperlink network with multi-hop questions, ensuring a realistic and challenging evaluation setting.

### 3.1 NIAH Testing for RAG: Retrieval, Needle, and Haystack

RAG is a popular context engineering strategy due to its simplicity and broad applicability. In a standard RAG pipeline, a retrieval strategy first fetches the top-$N$ documents deemed most relevant to a query. These documents, along with the query, form the input context for an LLM. To achieve high recall, the hyperparameter $N$ is often set to a value much larger than the number of ground-truth supporting documents for queries. This practice inevitably sacrifices precision and introduces challenging "near-miss" distractors that have high retrieval scores (Xu et al., 2024). The problem is exacerbated when a query requires logically combining information from multiple supporting documents, as in multi-hop question answering (QA). Our empirical studies in Sec. 4.1 demonstrate that a larger $N$ is required to achieve comparable retrieval recall for multi-hop questions.

Correspondingly, the requirement for a large $N$ inherently is prone to create the NIAH problem, assuming that perfect retrieval could be achieved with a sufficiently large $N$, or equivalently, a sufficiently large context size. Building on this observation, we formalize the NIAH problem from a RAG perspective: Let $\mathcal{D}$ be the document corpus. For any given query $q$, a set of ground-truth documents $\mathcal{N}_q \subset \mathcal{D}$ is required to answer it; we refer to this set as the **needle**. A retrieval strategy $\mathcal{R}$ scores and ranks all documents in $\mathcal{D}$ based on their predicted relevance to $q$.

To construct the **haystack** $\mathcal{H}_q^{\mathcal{R}}(S)$ for a target context size of $S$ tokens, we first include all needle documents from $\mathcal{N}_q$. We then fill the remaining token budget by adding the top-ranked distractors from $\mathcal{D} \setminus \mathcal{N}_q$. If including the final distractor would exceed the budget, we truncate that to fit. Finally, $\mathcal{H}_q^{\mathcal{R}}(S)$ is linearized into a sequence of documents $(d_1, \cdots, d_{|\mathcal{H}_q^{\mathcal{R}}(S)|})$ according to an ordering policy $\pi(q, \mathcal{R}, \mathcal{H}_q^{\mathcal{R}}(S))$ (e.g., by retrieval ranking), before being passed to the LLM. See Appendix B for the detailed prompts we use.

### 3.2 Assessing Heterogeneous Retrieval Strategies

**Retriever Strategy ($\mathcal{R}$) and Haystack Composition.** The above formulation highlights the central role of retrieval strategy ($\mathcal{R}$) in haystack engineering. Different retrieval strategies introduce distinct biases into the distractor composition, which consequently shape the reasoning challenge for LLMs according to the strategy's specific failure modes. Since no single method is universally optimal in terms of both effectiveness and efficiency, it is crucial to consider heterogeneous retrievers. To this end, HaystackCraft incorporates a broad spectrum of retrievers, including:

1. **Sparse (BM25)** (Robertson et al., 1994; Robertson & Zaragoza, 2009): A classical sparse retriever that measures lexical similarity.
2. **Dense (Qwen3-Embedding-0.6B)** (Zhang et al., 2025): A dense retriever that captures semantic similarity. We choose this model for its competitive retrieval performance on MMTEB (Enevoldsen et al., 2025), small size, and applicability to long documents.
3. **Hybrid (BM25 + Qwen3-Embedding-0.6B)**: A combination of the two using reciprocal rank fusion (Cormack et al., 2009; Microsoft, 2025), which is robust to differences in score magnitudes across retrievers. As sparse and dense retrievers are complementary, a hybrid of them often yields better performance in practice (Lee et al., 2023).

**Graph-Based Reranking for Multi-Hop QA.** Complex queries, such as those in multi-hop QA (Trivedi et al., 2022), require synthesizing information from multiple interconnected documents (the needle set $\mathcal{N}_q$). Standard retrievers score documents independently and thus overlook the relational structure among documents (e.g., hyperlinks or citations), limiting their ability to surface supporting chains. This frames the task as finding a "needle subgraph" within a larger "haystack graph". Graph structure provides powerful retrieval signals. For instance, PageRank (Page et al., 1999), a foundational algorithm for modern search engines, leverages this by considering a document structurally important if it is heavily referenced by other important documents. Building on this idea, we employ Personalized PageRank (PPR) (Haveliwala, 2002) reranking to study the impact of graph-based retrieval on distractor composition and downstream LLM performance. Specifically, after retrieving an initial candidate set with a base retriever, we perform PPR reranking seeded on the top-ranked documents to integrate structural information.

**Haystack Ordering ($\pi$).** LLMs exhibit strong positional biases due to autoregressive generation and positional encodings, and the order of documents can significantly impact their long-context

performance (Liu et al., 2024b; Xiao et al., 2024; Yang et al., 2025c). While prior NIAH benchmarks often randomize document order to account for this issue, practical RAG systems present documents in a retriever-ranked order. To bridge this simulation-to-reality gap, we evaluate both retriever-ranked descending ordering and random permutations. This dual approach allows us to assess LLM performance in a realistic RAG setting while also diagnosing the effects of positional bias.

## 3.3 DYNAMIC NIAH TESTING FOR AGENTIC CONTEXT ENGINEERING

Standard RAG can be ineffective when dealing with imperfect queries or complex tasks. User queries might be ambiguous or contain grammatical errors, which harm effective retrieval. Furthermore, standard RAG struggles with multi-hop queries, which are composed of logically interdependent subqueries. In this case, retrieving enough evidence requires answering earlier subqueries first. For instance, to answer "*What continent is the country encompassing Luahoko located in?*", a system must first find that Luahoko is in Tonga before a second retrieval for the continent of Tonga .

Agentic context engineering (Yao et al., 2023) can mitigate these limitations by transforming LLMs from passive retrieval consumers into proactive researchers. In such systems, LLMs can dynamically initiate further retrievals as needed (Trivedi et al., 2023; Jiang et al., 2023), refine queries to optimize retrieval quality (e.g., replacing the query above with "*What continent is Tonga located in?*") (Wang et al., 2023; Ma et al., 2023), and reflect on their past analyses (Shinn et al., 2023; Asai et al., 2024) until they can confidently draw a conclusion.

However, agentic context engineering introduces a new challenge: **LLMs themselves become a potential source of distraction**. Recent studies show that even advanced reasoning models struggle to recognize their own reasoning errors, often *reinforcing initial mistakes* rather than *correcting them* (Huang et al., 2024; He et al., 2025). Early-stage errors, such as noisy retrievals or flawed reasoning, can propagate and compound through LLMs' generation, leading to cascading failures or a gradual deviation from the original query's intent. While related issues have been previously observed (Laban et al., 2025), the interplay between wider context windows and deeper agentic iterations introduces a critical failure mode not captured by existing static NIAH benchmarks and multi-round benchmarks. This gap highlights a need for benchmarks that can test these integrated "wide and deep" long-context challenges. Existing static, single-round NIAH tests are insufficient, motivating our development of a dynamic, multi-round, and LLM-dependent test environment.

We perform an extension of our previous NIAH formulation in Section 3.1 for comparable results, simplicity, and controllability, while capturing key characteristics: multi-round retrieval, query refinement, and self-reflection. The process is iterative. We start with the original query $q^{(0)} = q$ and an empty LLM reasoning history $\mathcal{C}^{(0)} = ()$. At each round $t$, we use the latest query $q^{(t)}$ to construct the haystack $\mathcal{H}_{q^{(t)}}^{\mathcal{R}}(S)$. The LLM receives $q^{(t)}$, the history $\mathcal{C}^{(t)}$, and the ordered haystack. In intermediate rounds, the LLM outputs a refined query $q^{(t+1)}$ and its latest analysis $\mathcal{A}^{(t+1)}$, and we update the history $\mathcal{C}^{(t+1)} = (\mathcal{A}^{(1)}, \cdots, \mathcal{A}^{(t+1)})$. In the final round, the LLM outputs its answer.

We evaluate models in two dynamic settings, with detailed prompts provided in Appendix C:

- **Enforced Multi-Round**. We enforce models to perform a **fixed** number of reasoning rounds to measure their robustness against cascading errors.
- **Variable-Round**. We allow models to **decide when to stop**, testing their ability to balance iterative refinement against the risk of cascading errors.

## 3.4 CORPUS AND QA SAMPLES

To instantiate the static and dynamic NIAH tests introduced above, we need a networked corpus and QA dataset that support heterogeneous retrieval strategies and multi-hop reasoning. In this subsection, we detail our choice of corpus and QA samples that satisfy the need.

**Networked Corpus.** We ground our benchmark in the entire English Wikipedia hyperlink network. This choice is deliberate: Wikipedia is a dominant information source for retriever development and QA dataset curation and serves as a widely recognized proxy for general knowledge (Chen et al., 2017). It provides a centralized testbed for studying realistic haystack engineering with diverse

retrievers. Furthermore, its large scale and natural network structure, formed by in-text references (hyperlinks), make it an ideal corpus for studying graph-based retrieval. We process the 2025-04-04 Wikipedia dump using WikiExtractor (Attardi, 2015), resulting in a network that comprises $6,954,909$ articles interconnected by $97,442,472$ unique hyperlinks.

**Long Retrieval Unit.** We choose to use full articles as the retrieval unit, rather than smaller, broken chunks. This approach mirrors modern search engines, which return entire documents, and avoids fragmenting a document's logical flow as with common chunking practices in RAG. By preserving article integrity, we present a more realistic and demanding long-context reasoning challenge.

**QA Datasets.** We use two established datasets: Natural Questions (NQ) (Kwiatkowski et al., 2019) for single-hop questions and MuSiQue (Trivedi et al., 2022) for multi-hop questions. The multi-hop questions require reasoning over up to four supporting documents, presenting a challenge that motivates agentic context engineering. Both datasets are built on Wikipedia, providing a unified source for needles and distractors. We choose MuSiQue over alternatives (Yang et al., 2018; Ho et al., 2020) as it is specifically designed to be less susceptible to reasoning shortcuts. Since both datasets were curated on earlier Wikipedia versions, we manually filter all samples to ensure validity under our updated corpus, addressing issues like outdated knowledge and ambiguity. This yields a final set of 500 high-quality samples. Further details are available in Appendix E.

**Data Contamination Mitigation.** A critical concern in LLM evaluation is data contamination, where exposure to benchmark data during pretraining inflates performance (Sainz et al., 2023). While the models we evaluate have almost certainly been trained on versions of Wikipedia and even the QA datasets, our benchmark's design inherently mitigates this risk. The core task demands locating the "needle" within a long context of plausible, retriever-selected distractors—rather than simple fact recall. This challenge is amplified for our multi-hop questions, which require synthesizing information across multiple documents, a process robust to memorization. Furthermore, our use of a recent Wikipedia dump post-dates the training cutoffs of most current LLMs, minimizing data overlap. Our empirical results in Section 4.1 confirm this mitigation: all models show substantial performance degradation as context size increases, demonstrating that they are actively reasoning over the provided text, not merely recalling memorized answers.

## 4 EXPERIMENTS

### 4.1 NIAH WITH HETEROGENEOUS RETRIEVAL STRATEGIES

To investigate the impact of retrieval strategies, we evaluate 12 open-source and commercial long-context LLMs across input context sizes of $S \in \{8K, 16K, 32K, 64K, 128K\}$. Our selection spans both reasoning models—three Qwen3 variants (Yang et al., 2025b), Gemini 2.5 Flash-Lite, and o4-mini—and leading general-purpose models, including GPT-4.1 mini and the open-source Llama-3.1 (Dubey et al., 2024), Qwen2.5-1M (Yang et al., 2025a), and Gemma 3 (Kamath et al., 2025) families. Following the practice of MuSiQue, we use the F1 score as the QA metric. Our analysis in Appendix G confirms that multi-hop questions are less susceptible to data contamination, making them preferable for our study. To ensure a fair comparison, we standardize token counts using the Qwen2.5-1M tokenizer. For more LLM and retriever setup details, see Appendix F.

**Retrieval Effectiveness.** As a preliminary step, we first evaluate the effectiveness of the retrieval strategies to ensure they construct meaningful distractors. We measure both Recall @$N$, which quantifies the coverage of ground-truth supporting documents, and NDCG @$N$ (Järvelin & Kekäläinen, 2000; 2002), which additionally accounts for the ranking. We study the scaling behavior of retrieval by increasing $N$, the number of retrieved documents, which directly corresponds to increasing the context size in our NIAH setting.

Fig. 2 shows that retrieval performance for all methods improves as $N$ increases, justifying constructing longer contexts with more distractors. Among the base retrievers, the dense retriever (Qwen3-0.6B) consistently outperforms the sparse retriever (BM25) in both metrics, and combining them with a hybrid retriever further improves the performance. The retrieval effectiveness decreases as the question hop count (# supporting documents) increases, validating our claim in Sec. 3.1 that multi-hop questions necessitate a larger $N$, leading to more distractors. Graph-based reranking with PPR boosts all base retrievers in both coverage and ranking, especially for multi-hop questions. For

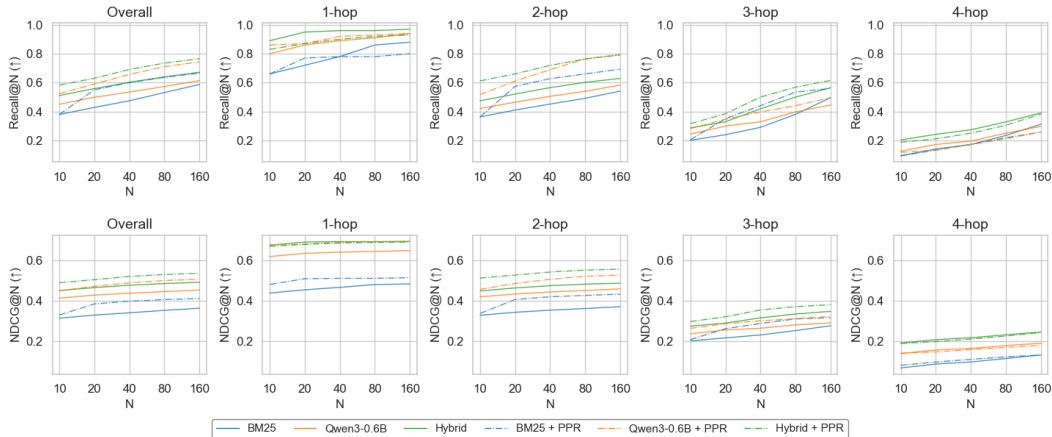

Figure 2: (1) Retrieval performance improves as # retrieved documents ($N$) increases. (2) Multi-hop questions pose larger retrieval challenges. (3) Reranking with PPR boosts performance, especially for 2-hop and 3-hop questions. See Appendix L for the raw numbers.

2-3-hop questions, PPR is highly effective as the supporting documents are typically located close to the top-ranked retrieved seeds in the graph. For 4-hop questions, the supporting documents can be much farther away, making long-range diffusion less targeted. As a result, PPR may miss some relevant documents or introduce false positives, leading to the small degradation observed.

**Impact of Retriever Strategy on NIAH Performance.** To study the overall impact of the retrieval strategy ($\mathcal{R}$) on haystack composition ($\mathcal{H}_q^{\mathcal{R}}(S)$) and ordering, we first employ retrieval ranking for haystack ordering ($\pi$). Fig. 3 presents the evaluation results. All LLMs, including advanced commercial and reasoning models, suffer significant performance degradation as context size increases to $128K$ tokens, regardless of the retrieval strategy. For 11 out of 12 cases, the dense retriever (Qwen3-0.6B) introduces more challenging distractors than the sparse retriever (BM25) at larger context sizes. However, combining them with a hybrid retriever does not necessarily introduce more challenging distractors.

**Impact of Graph-Based Retrieval.** Using PPR for graph-based reranking leads to significant NIAH performance improvement. By comparing the solid lines with the dashed lines in Fig. 3, we observe that for nearly every model and base retriever, the performance curve paired with PPR is noticeably higher, especially at context sizes of $64K$ and $128K$. This demonstrates that exploiting the relational structure among documents is a powerful method for mitigating distraction. For instance, an improvement of $44\%$ was observed for Llama-3.1-70B-Instruct with the hybrid retriever, highlighting how prioritizing structurally central documents can mitigate more harmful structurally isolated lexical and semantic distractors. Appendix I presents case studies for harmful distractor mitigation.

**Retrieval Effectiveness vs NIAH Performance.** Previous study by Jin et al. (2025) suggests that better retrievers introduce harder distractors for shorter-context reasoning and single-hop QA. Our study discloses a deeper insight, demonstrating that the interplay between the retriever mechanism and task nature plays a crucial role. While hybrid retriever substantially improves retrieval recall and ranking, it fails to introduce more challenging distractors. In contrast, graph-based reranking simultaneously improves retrieval effectiveness and mitigates harmful distractors. Our study highlights the critical role of retrieval strategy design in long-context engineering.

**Impact of Haystack Ordering.** To isolate the effect of haystack ordering ($\pi$), we compare the performance of retriever-ranked descending ordering against the average of three random permutations by visualizing the performance difference. The results in Fig. 4 reveal complex and highly model-dependent patterns. While some models, such as the Gemma-3 and Qwen2.5-1M families, derive a significant and growing benefit from retriever-ranked ordering as context size expands, others exhibit a more volatile, retriever-dependent, or even negative response. This finding carries a crucial implication: to faithfully assess a model's practical long-context utility in RAG systems, evaluations must mirror the canonical, retriever-ranked input. Furthermore, contrasting this setup with random permutations allows us to better understand the positional biases of individual models. Our ablation

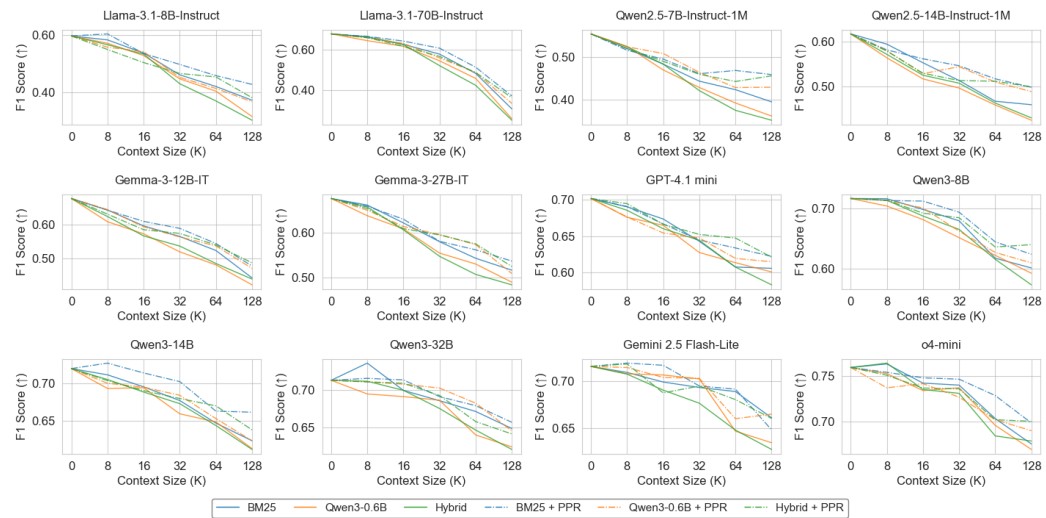

Figure 3: Impact of retrieval strategy on NIAH performance as context size increases. 0 stands for the case without distractors. All models experience a performance drop as context size increases. Graph-based reranking (dashed lines) consistently improves performance for larger context sizes. See Appendix M for the raw numbers.

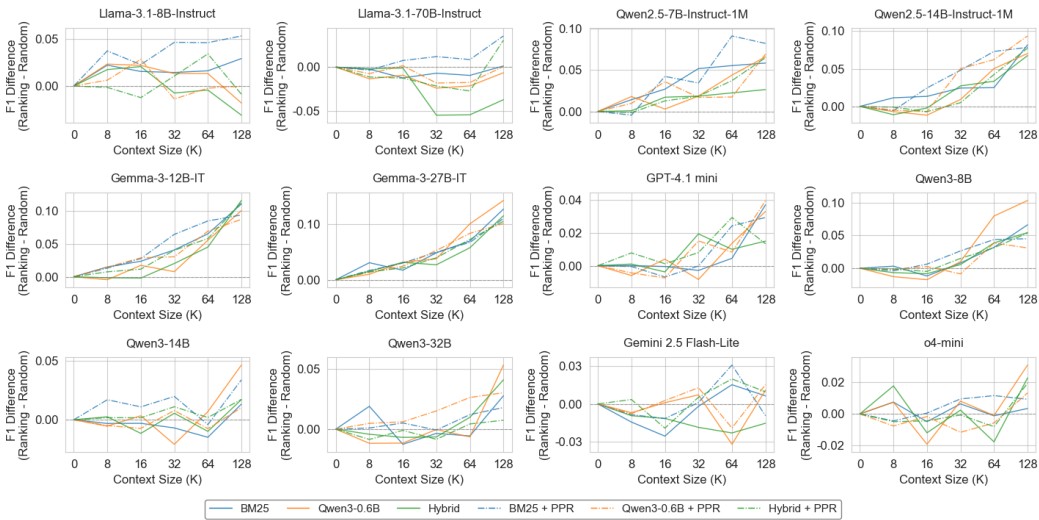

Figure 4: F1 score difference between retriever-ranked descending ordering and average over 3 random orderings. The ordering impact is highly model-dependent. The Gemma-3 and Qwen2.5-1M families derive a significant and growing benefit from retriever-ranked ordering as context size expands. See Appendix N for the raw NIAH performance numbers with random haystack orderings.

study in Appendix J further shows that models gaining the most from retriever-ranked descending ordering are also those most affected by the "lost in the middle" phenomenon (Liu et al., 2024b).

## 4.2 DYNAMIC NIAH

To assess the "wide and deep" challenges in agentic context engineering (Sec. 3.3), we perform dynamic NIAH evaluation with multi-round reasoning. We randomly choose 100 QA samples and evaluate eight LLMs, including state-of-the-art models Gemini 2.5 Pro and GPT-5. We exclusively use retriever-ranked haystack ordering, as this realistic setup ensures that LLM's query refinement is directly reflected in the context. Flawed refinements degrade the document ranking, posing a dy-

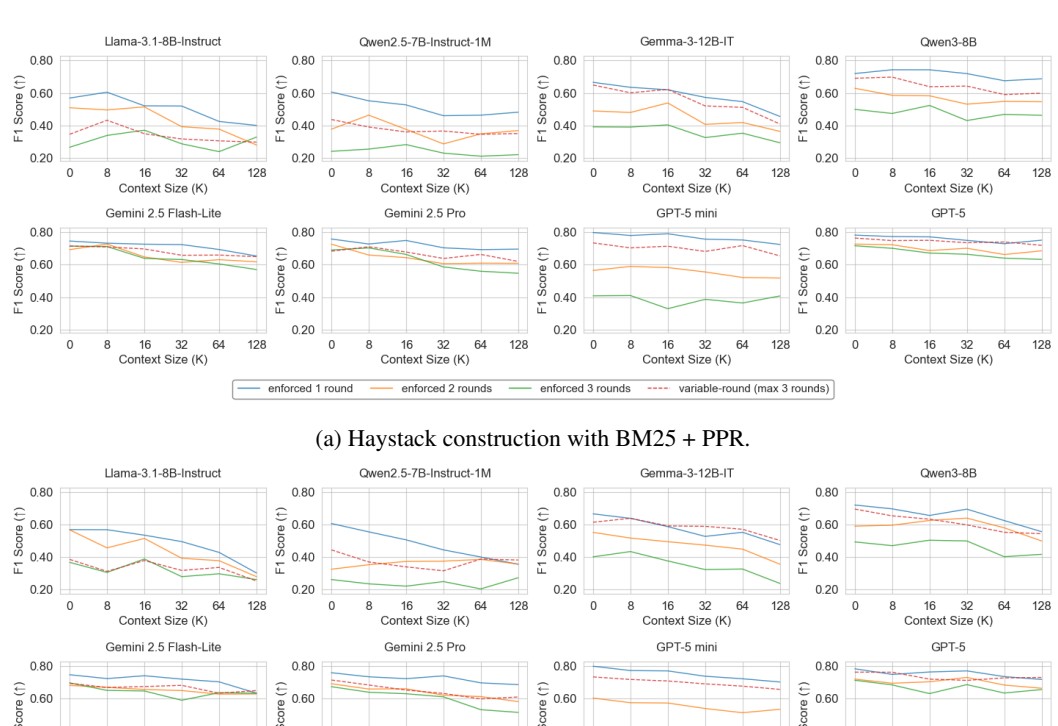

(a) Haystack construction with BM25 + PPR.

(b) Haystack construction with Qwen3-0.6B.

Figure 5: Dynamic NIAH performance. 0 stands for the case without distractors. (1) Enforced multi-round reasoning leads to performance drop. (2) Models are generally more robust to wider contexts than deeper reasoning. (3) Models fail to perform early stop properly (variable-round). For raw experiment numbers, see Appendix O.

namic challenge that faithfully evaluates a model's robustness to cascading errors. We consider two representative retrieval strategies: BM25 + PPR, a graph-based strategy effective at mitigating more harmful distractors, and Qwen3-0.6B, a dense retriever that introduces more challenging distractors.

**Enforced Multi-Round: More Rounds Amplify Errors.** We first evaluate model robustness by enforcing a constant number (2 or 3) of reasoning rounds. Fig. 5 shows that all models, including the most advanced Gemini 2.5 Pro and GPT-5, are vulnerable to cascading errors. Across retrieval strategies and context sizes, performance generally worsens with more rounds. Rather than mitigating distraction, additional iterations often amplify early mistakes or inject new noise. Interestingly, the degradation is not always monotonic with context size in multi-round settings, and more rounds can be more damaging than longer noisy contexts in a single pass. Crucially, static NIAH performance is not a reliable predictor of multi-round robustness: for instance, GPT-5 mini performs comparably to GPT-5 in the static setting but collapses under enforced multi-round reasoning, revealing weaker agentic robustness.

**Variable-Round: Self-Correction Is Difficult.** We further investigate if models can balance iterative refinement against cascading errors when allowed to stop early before exhausting three rounds. Table 20 and 21 demonstrate that all models understand the instruction for early stop and perform it for at least 10% cases. However, none of the models reliably improve upon their single-round performance. GPT-5 achieves the best relative performance but still fails to convert multi-round reasoning into sustained improvements.

**Representative Failure Patterns.** Appendix K presents representative failure cases. 1) Early-stage errors can propagate and compound through query refinement and summarization, leading to cascading failures that are hard to correct. 2) LLMs can deviate from the original query intent,

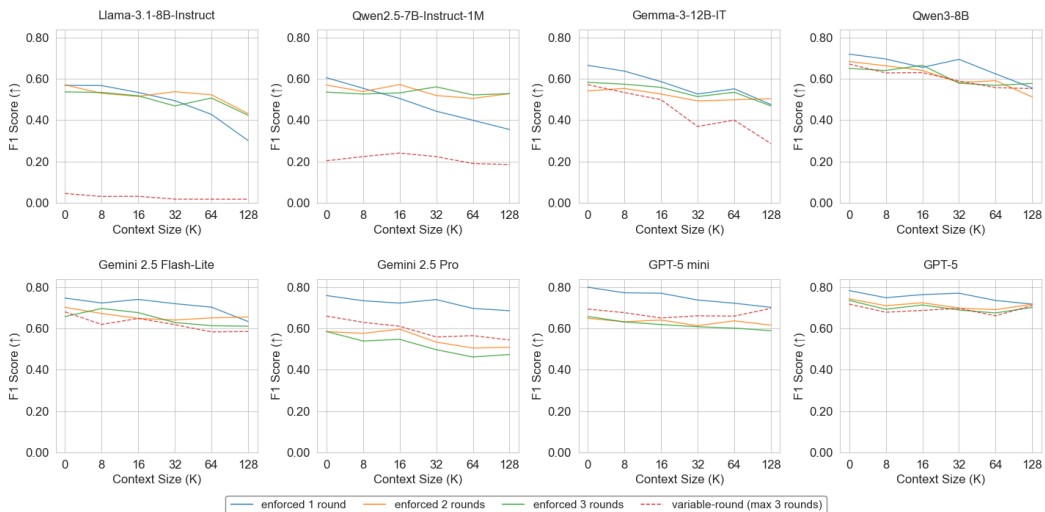

Figure 6: Ablation study for dynamic NIAH with a more advanced agent framework. 0 stands for the case without distractors. For raw experiment numbers, see Appendix R.

changing its nature or form. 3) Long-context challenges can still prevent LLMs from reasoning and retrieving relevant information.

**Implications.** Notably, the observations hold even when we constrain to single-hop questions, as detailed in Appendix Q. These findings reveal a consistent pattern: current LLMs are more robust to noisy long contexts than noisy reasoning iterations. In practice, practitioners should prioritize the use of a larger context window size ("width") over more reasoning iterations ("depth"). These results underscore the unsolved long-context challenges in agentic context engineering and establish HaystackCraft as a valuable testbed for measuring and advancing agentic robustness.

**Ablation: Sensitivity to Prompts and Workflow Designs**. Inspired by ReAct (Yao et al., 2023), we adopt a simple agent design for dynamic NIAH (Sec. 3.3), which provides a clean starting point that avoids conflating model behavior with heavily-engineered manual prompting. To verify whether cascading errors stem from the simplicity of our original workflow, we further evaluate a more advanced prompting design that: (1) explicitly highlights the original query, (2) provides the full history of paired retrieval queries and analyses from each round, and (3) encourages explicit reflection and correction of earlier findings. See Appendix D for the detailed prompts.

Fig. 6 shows that cascading errors persist for the strongest models (Gemini 2.5 and GPT-5), with advanced prompting yielding limited benefit. In contrast, weaker models (e.g., Llama-3.1-8B, Qwen2.5-7B) show notable improvements, sometimes even benefiting from additional rounds. However, the same models degrade sharply in the variable-round setting, where the added prompt complexity itself becomes a distractor and leads to failure in information gathering and aggregation.

These observations confirm that cascading errors are not merely an artifact of a simple workflow. A plausible explanation is that the latest reasoning models have already acquired the capabilities of reflection and query intention retaining through RL-based post-training (DeepSeek-AI et al., 2025), reducing the marginal advantage of further manual prompting. In contrast, weaker models benefit from explicit structure only when the prompt remains within their comprehension bandwidth; once the workflow becomes too complex, the prompting itself becomes an additional distractor.

## 5 CONCLUSION

Our benchmark, HaystackCraft, demonstrates that retrieval strategies critically shape distractor composition and ordering. Furthermore, our novel dynamic tests reveal that even state-of-the-art models like Gemini 2.5 Pro and GPT-5 remain vulnerable to cascading self-distractions and fail to self-correct. These findings highlight that robust agentic long-context reasoning is far from solved and establish HaystackCraft as a valuable testbed for measuring future progress.

**Reproducibility Statement.** We describe our dataset pre-processing steps in Sec. 3.4. For additional details, see Appendix B and C for prompt templates; see Appendix F for retriever and LLM hyperparameters. Furthermore, we include the raw experiment numbers associated with the figures in the Appendix. Finally, we include our code as supplementary material.

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

## A  LLM USAGE DISCLOSURE

We use Gemini 2.5 Pro, GPT-5, and Grok for writing enhancements, primarily to improve grammar and overall text flow. We also use their DeepResearch capabilities to retrieve related works for more contextualized discussions. All LLM outputs were reviewed and verified by the authors to ensure accuracy and avoid factual errors or hallucinations.

## B  NIAH PROMPT

---

**Input Prompt for NIAH Evaluation**

Read the following articles and answer the question below.

{ordered haystack}

What is the correct answer to this question: {question}

Format your response as follows: "The correct answer is (insert answer here)".

---

# C  DYNAMIC NIAH PROMPTS

## C.1  ENFORCED MULTI-ROUND

---

### Input Prompt for Intermediate Rounds in Enforced Multi-Round NIAH Evaluation

Read your previous analyses and the following articles. Analyze the question below.

Previous Analyses: {analyses}

Articles: {ordered haystack}

Question: {question}

Based on your previous analyses and the potentially new articles provided, summarize your findings related to the question and refine the question.

Format your response as follows:

Summary: (Summarize what you found in the articles that relates to the question, including any partial answers, relevant context, or gaps in information.)

Refined Question: (Copy the original question or replace it with a more specific question based on your findings.)

---

### Input Prompt for Final Round in Enforced Multi-Round NIAH Evaluation

Read your previous analyses and the following articles, and answer the question below.

Previous Analyses: {analyses}

Articles: {ordered haystack}

What is the correct answer to this question: {question}

Format your response as follows: "The correct answer is (insert answer here)".

---

## C.2  VARIABLE-ROUND

---

### Input Prompt for Variable-Round NIAH Evaluation

Read your previous analyses and the following articles. Analyze the question below.

Previous Analyses: {analyses}

Articles: {ordered haystack}

Question: {question}

Based on your previous analyses and the potentially new articles provided, decide if you are confident in answering the question or if you need additional information.

If you have complete information to fully answer the question, format your response as follows: "The correct answer is (insert answer here)".

If you need more information, format your response as follows:
Summary: (Summarize what you found in the articles that relates to the question, including any partial answers, relevant context, or gaps in information.)

Refined Question: (Copy the original question or replace it with a more specific question based on your findings.)

---

# D ALTERNATIVE PROMPTS FOR ABLATION STUDIES OF DYNAMIC NIAH

## D.1 ENFORCED MULTI-ROUND

---

**Input Prompt for Intermediate Rounds in Enforced Multi-Round NIAH Evaluation (Ablation)**

Read your previous analyses and the following latest retrieved articles. Refine and summarize your findings for answering the question.

Previous Analyses:
{round 1 retrieval query}
{round 1 analysis}
{round 2 retrieval query}
{· · · }

Latest Retrieved Articles: {ordered haystack}

Query Used for Retrieval: {retrieval query}

Original Question to Answer: {original question}

Based on the latest information, reflect on your earlier analyses. Update or correct them as needed. Summarize your current findings for answering the original question, and generate a refined query for the next retrieval round.

Format your response as follows:

Summary: (Summarize your findings for answering the original question, including any partial answers, relevant context, or gaps in information.)

Refined Query: (Generate the query for embedding-based retrieval in the next round based on your updated findings.)

---

**Input Prompt for Final Round in Enforced Multi-Round NIAH Evaluation (Ablation)**

Read your previous analyses and the following latest retrieved articles. Answer the question.

Previous Analyses:
{round 1 retrieval query}
{round 1 analysis}
{round 2 retrieval query}
{· · · }

Latest Retrieved Articles: {ordered haystack}

Query Used for Retrieval: {retrieval query}

What is the correct answer to this question: {original question}

Format your response as follows: "The correct answer is (insert answer here)".

---

## D.2  VARIABLE-ROUND

---

**Input Prompt for Variable-Round NIAH Evaluation (Ablation)**

Read your previous analyses and the following latest retrieved articles. Analyze the information for answering the question.

Previous Analyses:
{round 1 retrieval query}
{round 1 analysis}
{round 2 retrieval query}
{· · · }

Latest Retrieved Articles: {ordered haystack}

Query Used for Retrieval: {retrieval query}

Original Question to Answer: {original question}

Based on the latest information, decide if you are confident in answering the question or if you need additional information.

If you have complete information to fully answer the question, format your response as follows: "The correct answer is (insert answer here)".

Otherwise, based on the latest information, reflect on your earlier analyses. Update or correct them as needed. Summarize your current findings for answering the original question, and generate a refined query for the next retrieval round.

Format your response as follows:

Summary: (Summarize your findings for answering the original question, including any partial answers, relevant context, or gaps in information.)

Refined Query: (Generate the query for embedding-based retrieval in the next round based on your updated findings.)

---

# E  MORE DATASET DETAILS

In preparing the Wikipedia hyperlink network, we filter out empty and redirect pages.

Table 1 provides a dataset breakdown over hop count.

Table 1: Question breakdown over hop count.

| # hops | %    |
|--------|------|
| 1      | 20   |
| 2      | 58   |
| 3      | 15.6 |
| 4      | 6.4  |

# F  ADDITIONAL SETUP DETAILS

## F.1  LLM SETUP

For each LLM, we utilize the recommended inference hyperparameters as specified on its Hugging Face model card. These settings include sampling parameters like temperature, Top-P, Top-K, and Min-P, along with the "thinking budget" for thinking LLMs. All models considered in this work possess native long-context support for at least $128K$ tokens, with the exception of the Qwen3 models. To ensure the Qwen3 models could process a $128K$-token input and generate a $32K$-token output, we extend their context window to $164K$ tokens using YaRN (Peng et al., 2024).

## F.2  PPR SETUP

We perform a hyperparameter search for PPR per retriever using $10\%$ of the QA samples. For retrieval criteria, we adopt Normalized Discounted Cumulative Gain (NDCG) @ $10K$ (Järvelin &

Kekäläinen, 2000; 2002) for ranking ground truth supporting documents among the corpus. Table 2 presents the best hyperparameters for each retriever based on three random seeds.

Table 2: Retriever-specific PPR hyperparameters.

| Retriever | # Seed Documents | Damping Factor |
|---|---|---|
| BM25 | 10 | 0.5 |
| Qwen3-0.6B | 5 | 0.5 |
| Hybrid | 5 | 0.85 |

## G EVALUATION FOR DATA CONTAMINATION

To quantify data contamination, we evaluate LLM performance under two conditions: 1) without context, to test reliance on parametric knowledge, and 2) with ground-truth supporting documents. We measure F1 scores across an increasing number of the question hop count to assess how performance varies with reasoning complexity.

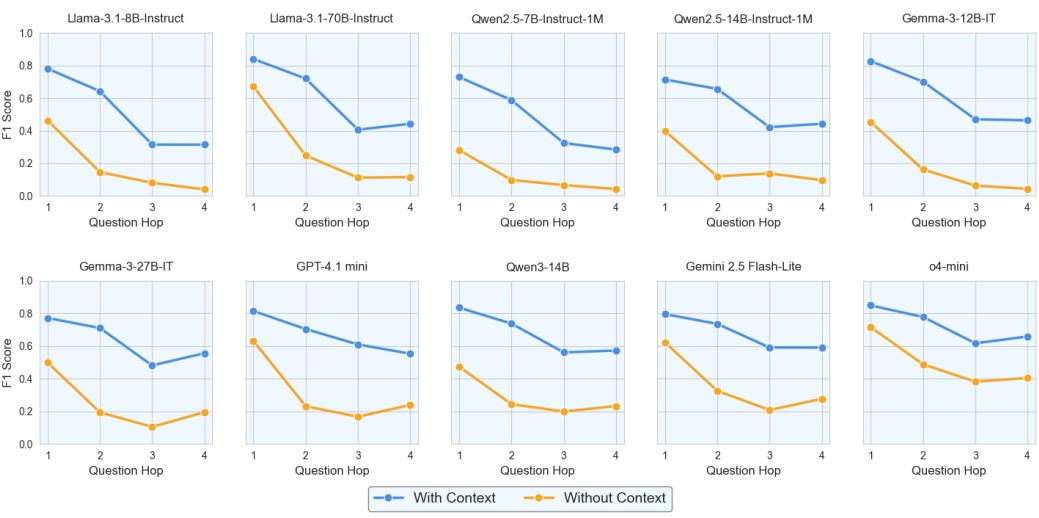

Figure 7: LLM performance with vs without context across question hop.

Fig. 7 presents the evaluation results.

- **Contamination is evident.** All models achieve non-zero F1 scores without context. This indicates a degree of data contamination.

- **Context is crucial.** Despite contamination, providing ground-truth documents substantially improves the performance of all models.

- **Complexity remains a challenge.** F1 scores generally decrease as the question hop count increases, even when context is provided. This also suggests that evaluation with multi-hop questions suffers less from data contamination.

## H BREAKDOWN OF STATIC NIAH PERFORMANCE OVER QUESTION HOP COUNT

Fig. 8 presents the break down of static NIAH performance over question hop count for Llama-3.1-8B-Instruct and Qwen2.5-7B-Instruct-1M.

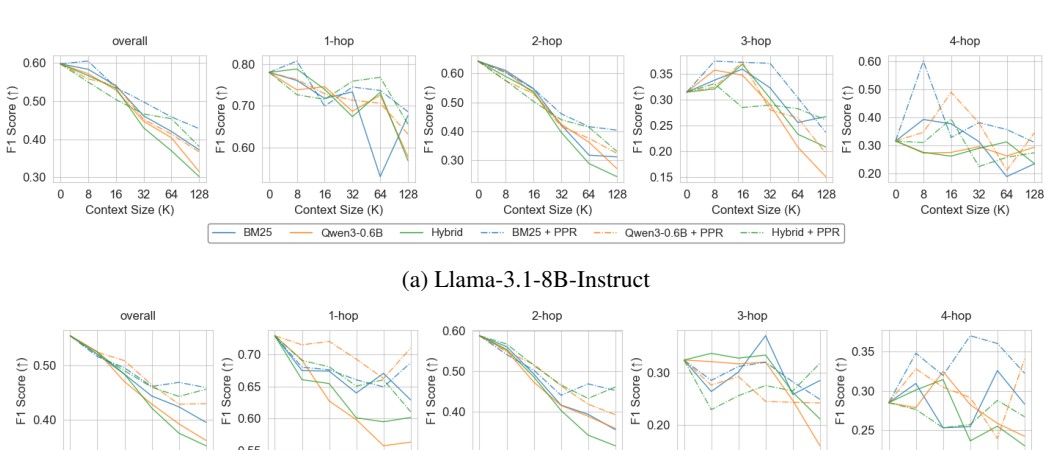

(a) Llama-3.1-8B-Instruct

(b) Qwen2.5-7B-Instruct-1M

Figure 8: NIAH performance break down over question hop count. 0 stands for the case without distractors. While larger fluctuations are observed for hop-specific subsets due to a smaller sample size, the qualitative patterns largely stay consistent.

## I CASE STUDIES FOR HARMFUL DISTRACTOR MITIGATION WITH PPR

> **What weekly publication in Timothy Pitkin's place of death is issued by John Kerry's university?**
>
> **Harmful distractor:** Initial retrieval introduces a harmful distractor article titled "**University of Florida Taser incident**", because it mentions "John Kerry", a university (University of Florida), and a publication issued by the university (The Independent Florida Alligator). A model then predicted "The Independent Florida Alligator" as the answer to the question.
>
> **PPR mitigation:** After PPR, the distractor article is no longer ranked high and thus avoided as it does not well connect to other top retrieval results, e.g., articles titled "Timothy Pitkin" and "New Haven, Connecticut".

> **In what year did the unification of the country containing the town of Sarikei happen?**
>
> **Harmful distractor:** Initial retrieval introduces harmful distractor articles like "Korean reunification", "German reunification", "Bulgarian unification", and "Unification of Moldova and Romania".
>
> **PPR mitigation:** The above distractors are eliminated after running PPR as they do not connect to the article "Sarikei".

> **What year was the band that performed 1000 Miles formed?**
>
> **Harmful distractor:** Initial retrieval introduces harmful distractor articles like "Band of Gypsys", "Bloodline (band)", "Sister Sadie (band)", and "Guillemots (band)". These articles also mention terms like "miles", causing models to make incorrect answers.
>
> **PPR mitigation:** The above distractors are eliminated after running PPR as they do not connect to the article "1000 Miles (Grinspoon song)".

> **Who is the Green Party member who studied at the largest employer in Sonny Berman's place of birth?**
>
> **Harmful distractor:** Initial retrieval introduces a harmful distractor article "Ricardo Menéndez March", which mentions terms like "Green Party" and "employ", leading to incorrect model responses.
>
> **PPR mitigation:** The above distractor is eliminated after running PPR as it does not connect to the article "Sonny Berman".

> **What year was the end of the company which made the Hudson Commodore?**
>
> **Harmful distractor:** Initial retrieval introduces a harmful distractor article "Holden", which mentions terms "year", "end", "company", "Hudson", and "Commodore". Models then incorrectly treated the end year of "Holden" as the answer to the question.
>
> **PPR mitigation:** The above distractor is eliminated after running PPR as it does not connect to the article "Hudson Commodore".

## J   ABLATION STUDY FOR HAYSTACK ORDERING

The "lost-in-the-middle" phenomenon refers to the significant degradation in LLM performance when relevant information appears in the middle of a long context (Liu et al., 2024b). To examine its potential connection to our previous observations regarding the impact of haystack orderings, we conduct an additional ablation study using two representative models, Gemma-3-12B-IT and Gemini 2.5 Flash, on 128K-token contexts constructed with the hybrid retriever.

We evaluate four haystack orderings:

- Retriever-ranked descending
- Retriever-ranked ascending
- Average over three random permutations
- Middle, where the ground-truth supporting documents are manually placed at the center of the context.

Table 3: Ablation study for haystack ordering.

| Model | Descending | Ascending | Random | Middle |
|---|---|---|---|---|
| Gemma-3-12B-IT | **44.1** | 39.97 | 32.51 | 29.36 |
| Gemini 2.5 Flash | 63.28 | **65.44** | 64.06 | 65.31 |

As shown in Tab. 3, Gemma-3-12B-IT shows substantial sensitivity to ordering. Both descending and ascending rankings outperform the random and middle placements, and the middle condition yields the worst results, consistent with the "lost in the middle" effect. In contrast, Gemini 2.5 Flash remains robust across all orderings with minimal variation. These results clarify the model-specific pattern noted in Fig. 4. Models that benefit most from retriever-ranked descending ordering are also the ones that suffer most from the "lost-in-the-middle" issue.

## K   CASE STUDIES FOR FAILURE PATTERNS IN DYNAMIC NIAH

> **Gemma-3-12B-IT propagates its reasoning errors via query refinement and summarization.**
>
> [Original query]: What body of water is by the place John Dury died?
>
> [First-round response]:
>
> Summary: The provided text discusses John Dury and his life, focusing on his religious and intellectual work. It states that he died in Edinburgh. Edinburgh is located on the Firth of Forth. Therefore, John Dury died near the Firth of Forth.   // Wrong. He was born in Edinburgh.
>
> Refined Question: What body of water is located near Edinburgh, where John Dury died?   // The mistake got consolidated and will inflate the ranking of distractor documents.
>
> [Second-round response]:
>
> Summary: The provided articles offer several pieces of information related to John Dury and the geography of Edinburgh. Firstly, the article on Edinburgh states it's located "near the Firth of Forth". Secondly, the article on John Dury mentions that he died in Edinburgh, and Edinburgh is located on the Firth of Forth. Therefore, John Dury died near the Firth of Forth. The articles do not offer any other specific bodies of water near Edinburgh.
>
> Refined Question: What body of water is Edinburgh situated on, in which John Dury died?   // The LLM cannot correct consolidated early mistakes.
>
> [Final answer prediction]: The correct answer is the Firth of Forth.   // The final answer is then naturally wrong.

---

**GPT-5 deviated from the original query intent.**

[Original query]: What year saw the formation of the Brazilian republic?

[Refined query after the first round]: In what year (and on what date) was the Brazilian Republic proclaimed?  // GPT-5 added "on what date"

[Refined query after the second round]: In what year, and on what date, was the Brazilian Republic proclaimed? (Answer: 15 November 1889)  // GPT-5 reinforced the earlier query deviation.

[Final answer prediction]: The correct answer is 15 November 1889.  // The correct answer should be just 1889.

---

**GPT-5 mini changed open questions to yes/no questions.**

[Original query]: What country is the member of the band that performed Visions of Eden a citizen of?

[Refined query after the first round]: Which country is David DeFeis (the Virgin Steele member who performed on Visions of Eden) a citizen of?

[Refined query after the second round]: Is David DeFeis a citizen of the United States? If so, can you cite a source that explicitly confirms his country of citizenship (e.g., place of birth, official biography or other primary documentation)?  // The question got changed to an yes/no question.

[Final answer prediction]: Yes.  // GPT-5 mini even failed to answer its own second question, despite the deviation.

---

**GPT-5 mini changed open questions to multiple-choice questions.**

[Original query]: What year did unification of the country Mintu is located happen?

[Refined query after the first round]: Do you mean the year Sarawak became part of the federation of Malaysia (i.e., when Malaya, North Borneo/Sabah, Sarawak and Singapore united) — which was 1963 — or do you mean the earlier independence of the Federation of Malaya in 1957?

[Refined query after the second round]: Do you mean (A) the year Sarawak (and thus Mintu) joined the Federation of Malaysia — Malaysia Day, 16 September 1963 — or (B) the earlier independence of the Federation of Malaya — Hari Merdeka, 31 August 1957?  // The question got changed to a multiple-choice question.

[Final answer prediction]: A.

---

**Gemini 2.5 Pro fails due to long-context challenges.**

[Original query]: What is the nickname of the state where John Kerry's VP pick was a senator?

[First-round response]:

Summary: The articles identify John Kerry's vice presidential running mate in the 2004 presidential election as John Edwards. At the time of his selection, Edwards was a U.S. Senator from North Carolina. The article titled "North Carolina" provides two slogans used on the state's license plates: "First in Flight" and "First in Freedom". While the article also refers to "Tar heel voters", it does not explicitly state that "The Tar Heel State" is the official nickname.

Refined Question: What is the nickname of the state where John Kerry's VP pick was a senator?  // Gemini 2.5 Pro simply fails in identifying relevant text about "Old North State". It also does not update the query properly by incorporating partial evidence.

[Second-round response]: Summary: The articles confirm that John Kerry's vice presidential running mate in the 2004 presidential election was John Edwards. Multiple articles state that Edwards was a U.S. Senator from North Carolina at the time of his selection. The article on North Carolina mentions two slogans used on the state's license plates, "First in Flight" and "First in Freedom", but none of the provided texts explicitly identify the state's official nickname.

Refined Question: What is the nickname of the state where John Kerry's VP pick was a senator?

[Final answer prediction]: Based on the provided articles, the correct answer is that the texts do not explicitly state an official nickname for North Carolina.

## L  RAW EXPERIMENT RESULTS FOR RETRIEVAL EVALUATION

See Table 4 and Table 5.

Table 4: Recall@$N$ of retrieval strategies for coverage evaluation ($\times 10^{-2}$, $\uparrow$), with a breakdown over question hop. We present the results in a way that allows comparing the impact of using PPR or not, and we highlight the better results.

| Base Retriever | Hop | PPR | @10 | @20 | @40 | @80 | @160 |
|---|---|---|---|---|---|---|---|
| BM25 | Overall | | 37.93 | 42.83 | 47.43 | 53.13 | 58.73 |
| | | ✓ | **38** | **55.02** | **59.97** | **63.63** | **66.58** |
| BM25 | 1-hop | | **66** | 72 | **78** | **86** | **88** |
| | | ✓ | **66** | **77** | **78** | 78 | 80 |
| BM25 | 2-hop | | **36.21** | 41.03 | 45.17 | 49.14 | 54.14 |
| | | ✓ | **36.21** | **57.41** | **62.76** | **66.03** | **69.31** |
| BM25 | 3-hop | | 20.09 | 23.93 | 29.06 | 38.03 | 49.57 |
| | | ✓ | **20.51** | **35.04** | **44.02** | **53.42** | **55.98** |
| BM25 | 4-hop | | **9.38** | **14.06** | **17.19** | **23.44** | **31.25** |
| | | ✓ | **9.38** | 13.28 | **17.19** | 21.88 | 25.78 |
| Qwen3-0.6B | Overall | | 45 | 49.87 | 53.48 | 57.3 | 61.43 |
| | | ✓ | **52.35** | **59.05** | **65.7** | **71.12** | **74.28** |
| Qwen3-0.6B | 1-hop | | 80 | 86 | 89 | 91 | **94** |
| | | ✓ | **86** | **87** | **92** | **93** | **94** |
| Qwen3-0.6B | 2-hop | | 42.07 | 46.38 | 50.52 | 53.97 | 58.28 |
| | | ✓ | **51.72** | **61.03** | **68.97** | **76.38** | **79.48** |
| Qwen3-0.6B | 3-hop | | 24.36 | 29.91 | 32.91 | 39.74 | 44.44 |
| | | ✓ | **28.21** | **34.62** | **39.74** | **44.02** | **49.57** |
| Qwen3-0.6B | 4-hop | | **12.5** | **17.19** | **19.53** | **25** | **29.69** |
| | | ✓ | 11.72 | 13.28 | 17.19 | 21.09 | 25.78 |
| Hybrid | Overall | | 51.07 | 55.88 | 60.28 | 64 | 67.2 |
| | | ✓ | **58.23** | **63.05** | **69.1** | **73.52** | **76.55** |
| Hybrid | 1-hop | | **89** | **95** | **96** | **96** | **97** |
| | | ✓ | 83 | 87 | 90 | 92 | 93 |
| Hybrid | 2-hop | | 47.41 | 52.07 | 56.55 | 60.17 | 62.93 |
| | | ✓ | **61.21** | **66.03** | **71.9** | **76.38** | **79.14** |
| Hybrid | 3-hop | | 28.63 | 32.91 | 41.88 | 50 | 56.41 |
| | | ✓ | **31.62** | **38.46** | **50** | **56.84** | **61.54** |
| Hybrid | 4-hop | | **20.31** | **24.22** | **27.34** | **32.81** | **39.06** |
| | | ✓ | 18.75 | 21.09 | 25 | 30.47 | 38.28 |

# M  RAW EXPERIMENT RESULTS FOR STATIC NIAH WITH RETRIEVAL-RANKED HAYSTACK ORDERING

- BM25: Table 6
- Qwen3-0.6B: Table 7
- Hybrid: Table 8
- BM25 + PPR: Table 9
- Qwen3-0.6B + PPR: Table 10
- Hybrid + PPR: Table 11

Table 5: NDCG@$N$ of retrieval strategies for ranking evaluation ($\times 10^{-2}, \uparrow$), with a breakdown over question hop. We present the results in a way that allows comparing the impact of using PPR or not, and we highlight the better results.

| Base Retriever | Hop | PPR | @10 | @20 | @40 | @80 | @160 |
|---|---|---|---|---|---|---|---|
| BM25 | Overall | | 31.31 | 32.83 | 33.96 | 35.16 | 36.22 |
| | | ✓ | **32.86** | **38.35** | **39.66** | **40.49** | **41.03** |
| BM25 | 1-hop | | 43.71 | 45.3 | 46.5 | 47.9 | 48.2 |
| | | ✓ | **47.94** | **50.77** | **50.99** | **50.99** | **51.27** |
| BM25 | 2-hop | | 32.74 | 34.22 | 35.25 | 36.07 | 36.97 |
| | | ✓ | **33.64** | **40.52** | **41.87** | **42.55** | **43.14** |
| BM25 | 3-hop | | 20.14 | 21.58 | 23.03 | 25.17 | 27.56 |
| | | ✓ | **20.76** | **26.13** | **28.7** | **30.95** | **31.49** |
| BM25 | 4-hop | | 6.76 | 8.69 | 9.71 | 11.4 | **13.18** |
| | | ✓ | **8.06** | **9.71** | **10.98** | **12.24** | 13.12 |
| Qwen3-0.6B | Overall | | 41.18 | 42.7 | 43.6 | 44.43 | 45.2 |
| | | ✓ | **44.91** | **47.04** | **48.72** | **49.89** | **50.49** |
| Qwen3-0.6B | 1-hop | | 61.72 | 63.3 | 63.89 | 64.22 | 64.65 |
| | | ✓ | **67.53** | **67.79** | **68.79** | **68.96** | **69.11** |
| Qwen3-0.6B | 2-hop | | 41.86 | 43.2 | 44.24 | 44.96 | 45.75 |
| | | ✓ | **45.53** | **48.42** | **50.46** | **52.02** | **52.58** |
| Qwen3-0.6B | 3-hop | | 23.58 | 25.51 | 26.35 | 28.01 | 28.97 |
| | | ✓ | **26.25** | **28.57** | **30.03** | **31.07** | **32.2** |
| Qwen3-0.6B | 4-hop | | 13.8 | **15.63** | **16.35** | **17.83** | **18.92** |
| | | ✓ | **14.09** | 14.67 | 15.86 | 16.87 | 17.93 |
| Hybrid | Overall | | 44.91 | 46.41 | 47.57 | 48.39 | 49 |
| | | ✓ | **48.81** | **50.3** | **51.89** | **52.86** | **53.45** |
| Hybrid | 1-hop | | **67.33** | **68.87** | **69.06** | **69.06** | **69.21** |
| | | ✓ | 66.75 | 67.74 | 68.36 | 68.69 | 68.84 |
| Hybrid | 2-hop | | 44.75 | 46.21 | 47.34 | 48.1 | 48.59 |
| | | ✓ | **51.09** | **52.58** | **54.07** | **55.03** | **55.53** |
| Hybrid | 3-hop | | 27.37 | 28.89 | 31.46 | 33.37 | 34.68 |
| | | ✓ | **29.62** | **32.01** | **35.36** | **37** | **37.95** |
| Hybrid | 4-hop | | **19.06** | **20.71** | **21.66** | **23.1** | **24.51** |
| | | ✓ | 18.82 | 19.76 | 21 | 22.47 | 24.24 |

## N  RAW EXPERIMENT RESULTS FOR STATIC NIAH AVERAGED OVER THREE RANDOM HAYSTACK ORDERINGS

- BM25: Table 12
- Qwen3-0.6B: Table 13
- Hybrid: Table 14
- BM25 + PPR: Table 15
- Qwen3-0.6B + PPR: Table 16
- Hybrid + PPR: Table 17

Table 6: Static NIAH performance in F1 score $(\times 10^{-2}, \uparrow)$ using BM25 for haystack construction, where retriever-ranked haystack ordering is used. 0 stands for the case without distractors.

| Context Size (# Tokens) | 0 | 8K | 16K | 32K | 64K | 128K |
|---|---|---|---|---|---|---|
| Llama-3.1-8B-Instruct | 59.8 | 58.41 | 53.81 | 46.16 | 42.06 | 37.24 |
| Llama-3.1-70B-Instruct | 67.7 | 66.28 | 62.56 | 57.72 | 48.2 | 30.71 |
| Qwen2.5-7B-Instruct-1M | 55.56 | 52.12 | 48.42 | 44.39 | 42.38 | 39.5 |
| Qwen2.5-14B-Instruct-1M | 61.76 | 59.46 | 55.28 | 51.3 | 46.65 | 45.87 |
| Gemma-3-12B-IT | 67.49 | 64.15 | 59.54 | 56.47 | 52.4 | 44.45 |
| Gemma-3-27B-IT | 67.71 | 66.24 | 62.25 | 58.05 | 54.34 | 51.68 |
| GPT-4.1 mini | 70.19 | 69.05 | 67.36 | 64.28 | 60.73 | 60.55 |
| Qwen3-8B | 71.66 | 71.62 | 69.84 | 67.94 | 61.72 | 60.1 |
| Qwen3-14B | 71.9 | 71.08 | 69.5 | 67.7 | 64.74 | 62.42 |
| Qwen3-32B | 71.32 | 73.62 | 69.97 | 68.59 | 67.15 | 64.87 |
| Gemini 2.5 Flash-Lite | 71.6 | 70.96 | 69.94 | 69.36 | 68.93 | 66.14 |
| o4-mini | 75.95 | 76.34 | 74.24 | 74.02 | 70.36 | 67.58 |

Table 7: Static NIAH performance in F1 score $(\times 10^{-2}, \uparrow)$ using Qwen3-0.6B for haystack construction, where retriever-ranked haystack ordering is used. 0 stands for the case without distractors.

| Context Size (# Tokens) | 0 | 8K | 16K | 32K | 64K | 128K |
|---|---|---|---|---|---|---|
| Llama-3.1-8B-Instruct | 59.8 | 57.22 | 52.92 | 44.74 | 40.42 | 31.51 |
| Llama-3.1-70B-Instruct | 67.7 | 64.36 | 61.6 | 54.42 | 45.73 | 25.85 |
| Qwen2.5-7B-Instruct-1M | 55.56 | 52.59 | 47.04 | 42.9 | 39.23 | 36.14 |
| Qwen2.5-14B-Instruct-1M | 61.76 | 56.4 | 51.66 | 49.61 | 45.8 | 42.39 |
| Gemma-3-12B-IT | 67.49 | 60.75 | 57.24 | 52.01 | 48.21 | 42.43 |
| Gemma-3-27B-IT | 67.71 | 63.86 | 60.92 | 55.5 | 53.05 | 48.98 |
| GPT-4.1 mini | 70.19 | 67.6 | 66.49 | 62.73 | 61.33 | 60.05 |
| Qwen3-8B | 71.66 | 70.41 | 68.15 | 65.14 | 62.2 | 59.22 |
| Qwen3-14B | 71.9 | 69.29 | 69.38 | 65.96 | 64.89 | 61.39 |
| Qwen3-32B | 71.32 | 69.49 | 69.14 | 68.65 | 64.01 | 62.41 |
| Gemini 2.5 Flash-Lite | 71.6 | 70.81 | 70.69 | 70.27 | 64.72 | 63.49 |
| o4-mini | 75.95 | 75.28 | 73.42 | 73.73 | 69.62 | 66.98 |

## O  RAW EXPERIMENT RESULTS FOR DYNAMIC NIAH

See Table 18 and Table 19.

## P  ANALYSIS OF EARLY STOP IN VARIABLE-ROUND DYNAMIC NIAH

Table 20 and Table 21 present the % cases where models perform an early stop.

## Q  DYNAMIC NIAH RESULTS FOR SINGLE-HOP QUESTIONS

See Table 22 for the raw experiment numbers and Fig. 9 for the visualization. Enforcing more rounds consistently degrades performance across models, even when we constrain to easier single-hop questions. In contrast, increasing context size alone is generally less harmful than increasing the number of rounds. Strong models such as GPT-5 and Gemini 2.5 Pro likewise fail to convert additional rounds into gains and often underperform their single-round results.

## R  RAW EXPERIMENT RESULTS FOR DYNAMIC NIAH (ABLATION)

See Table 23.

Table 8: Static NIAH performance in F1 score ($\times 10^{-2}, \uparrow$) using hybrid retriever for haystack construction, where retriever-ranked haystack ordering is used. 0 stands for the case without distractors.

| Context Size (# Tokens) | 0 | 8K | 16K | 32K | 64K | 128K |
|---|---|---|---|---|---|---|
| Llama-3.1-8B-Instruct | 59.8 | 56.71 | 53.41 | 43.02 | 36.99 | 30.22 |
| Llama-3.1-70B-Instruct | 67.7 | 66.09 | 62.44 | 52.04 | 42.33 | 25.11 |
| Qwen2.5-7B-Instruct-1M | 55.56 | 52.54 | 48.42 | 42.16 | 37.5 | 35.16 |
| Qwen2.5-14B-Instruct-1M | 61.76 | 57.17 | 52.48 | 50.7 | 46.24 | 42.91 |
| Gemma-3-12B-IT | 67.49 | 62.26 | 56.63 | 53.68 | 48.62 | 44.1 |
| Gemma-3-27B-IT | 67.71 | 65.79 | 60.7 | 54.78 | 50.73 | 48.4 |
| GPT-4.1 mini | 70.19 | 68.56 | 66.02 | 64.42 | 60.72 | 58.27 |
| Qwen3-8B | 71.66 | 71.36 | 68.74 | 66.52 | 61.54 | 57.29 |
| Qwen3-14B | 71.9 | 70.46 | 68.82 | 67.29 | 64.42 | 61.28 |
| Qwen3-32B | 71.32 | 71.19 | 69.98 | 67.55 | 64.66 | 62.07 |
| Gemini 2.5 Flash-Lite | 71.6 | 70.78 | 69.08 | 67.69 | 64.83 | 62.78 |
| o4-mini | 75.95 | 76.43 | 73.54 | 73.12 | 68.46 | 67.9 |
| GPT-5 | 77.28 | 77.21 | 75.37 | 75.15 | 74.99 | 73.76 |

Table 9: Static NIAH performance in F1 score ($\times 10^{-2}, \uparrow$) using BM25 + PPR for haystack construction, where retriever-ranked haystack ordering is used. 0 stands for the case without distractors.

| Context Size (# Tokens) | 0 | 8K | 16K | 32K | 64K | 128K |
|---|---|---|---|---|---|---|
| Llama-3.1-8B-Instruct | 59.8 | 60.52 | 53.63 | 49.81 | 45.87 | 42.8 |
| Llama-3.1-70B-Instruct | 67.7 | 66.49 | 64.14 | 60.66 | 51.26 | 37.08 |
| Qwen2.5-7B-Instruct-1M | 55.56 | 51.69 | 49.6 | 46.16 | 46.91 | 45.95 |
| Qwen2.5-14B-Instruct-1M | 61.76 | 58.17 | 56.17 | 54.61 | 51.68 | 49.78 |
| Gemma-3-12B-IT | 67.49 | 64.11 | 60.85 | 58.82 | 54.43 | 47.89 |
| Gemma-3-27B-IT | 67.71 | 65.98 | 63.08 | 58.13 | 56.23 | 53.71 |
| GPT-4.1 mini | 70.19 | 69.07 | 66.67 | 64.46 | 63.36 | 62.2 |
| Qwen3-8B | 71.66 | 71.32 | 71.23 | 69.38 | 64.44 | 62.37 |
| Qwen3-14B | 71.9 | 72.62 | 71.33 | 70.19 | 66.31 | 66.17 |
| Qwen3-32B | 71.32 | 71.58 | 71.37 | 69.13 | 67.93 | 65.7 |
| Gemini 2.5 Flash-Lite | 71.6 | 71.95 | 71.72 | 69.52 | 69.18 | 64.86 |
| o4-mini | 75.95 | 75.42 | 74.82 | 74.67 | 72.88 | 69.88 |

## S  IMPLEMENTATION DETAILS

We employ vLLM for LLM inference (Kwon et al., 2023).

Table 10: Static NIAH performance in F1 score $(\times 10^{-2}, \uparrow)$ using Qwen3-0.6B + PPR for haystack construction, where retriever-ranked haystack ordering is used. 0 stands for the case without distractors.

| Context Size (# Tokens) | 0 | 8K | 16K | 32K | 64K | 128K |
|---|---|---|---|---|---|---|
| Llama-3.1-8B-Instruct | 59.8 | 55.85 | 54.26 | 45.25 | 41.37 | 36.69 |
| Llama-3.1-70B-Instruct | 67.7 | 65.75 | 63.01 | 55.82 | 48.65 | 33.42 |
| Qwen2.5-7B-Instruct-1M | 55.56 | 52.51 | 50.93 | 46.47 | 42.88 | 42.96 |
| Qwen2.5-14B-Instruct-1M | 61.76 | 57.95 | 52.83 | 54.41 | 50.96 | 48.82 |
| Gemma-3-12B-IT | 67.49 | 64.25 | 59.27 | 56.33 | 53.72 | 47.2 |
| Gemma-3-27B-IT | 67.71 | 65.08 | 61.46 | 59.64 | 57.39 | 50.93 |
| GPT-4.1 mini | 70.19 | 67.64 | 65.43 | 64.74 | 61.92 | 61.45 |
| Qwen3-8B | 71.66 | 71.35 | 70.08 | 66.23 | 62.68 | 60.94 |
| Qwen3-14B | 71.9 | 69.97 | 69.46 | 68.39 | 65.33 | 62.42 |
| Qwen3-32B | 71.32 | 71.11 | 70.8 | 70.28 | 68.26 | 64.55 |
| Gemini 2.5 Flash-Lite | 71.6 | 71.48 | 70.43 | 70.33 | 66.05 | 66.53 |
| o4-mini | 75.95 | 73.73 | 74.18 | 72.7 | 70.24 | 69.03 |

Table 11: Static NIAH performance in F1 score $(\times 10^{-2}, \uparrow)$ using hybrid + PPR retriever for haystack construction, where retriever-ranked haystack ordering is used. 0 stands for the case without distractors.

| Context Size (# Tokens) | 0 | 8K | 16K | 32K | 64K | 128K |
|---|---|---|---|---|---|---|
| Llama-3.1-8B-Instruct | 59.8 | 55.04 | 50.45 | 46.64 | 45.41 | 38.11 |
| Llama-3.1-70B-Instruct | 67.7 | 65.85 | 61.56 | 56.6 | 48.64 | 36.22 |
| Qwen2.5-7B-Instruct-1M | 55.56 | 52.06 | 49.07 | 46.02 | 44.31 | 45.65 |
| Qwen2.5-14B-Instruct-1M | 61.76 | 57.92 | 52.93 | 51.29 | 51.14 | 49.88 |
| Gemma-3-12B-IT | 67.49 | 62.95 | 58.43 | 57.35 | 54 | 48.8 |
| Gemma-3-27B-IT | 67.71 | 65.45 | 60.97 | 59.56 | 57.55 | 52.51 |
| GPT-4.1 mini | 70.19 | 69.48 | 66.63 | 65.23 | 64.73 | 62.09 |
| Qwen3-8B | 71.66 | 71.37 | 69.22 | 68.45 | 63.59 | 63.98 |
| Qwen3-14B | 71.9 | 70.32 | 69 | 67.97 | 67 | 63.85 |
| Qwen3-32B | 71.32 | 71.11 | 70.93 | 69.29 | 65.8 | 64.14 |
| Gemini 2.5 Flash-Lite | 71.6 | 71.86 | 68.79 | 69.51 | 68.07 | 66.07 |
| o4-mini | 75.95 | 75.1 | 73.7 | 73.61 | 70.23 | 70.06 |
| GPT-5 | 77.28 | 77.15 | 77.04 | 76.12 | 73.83 | 73.64 |

Table 12: Static NIAH performance in F1 score $(\times 10^{-2}, \uparrow)$ using BM25 for haystack construction, where we average the results over three random Haystack orderings. 0 stands for the case without distractors.

| Context Size (# Tokens) | 0 | 8K | 16K | 32K | 64K | 128K |
|---|---|---|---|---|---|---|
| Llama-3.1-8B-Instruct | 59.8 | $56.18 \pm 0.85$ | $52.25 \pm 1.21$ | $44.71 \pm 0.86$ | $40.43 \pm 1.78$ | $34.3 \pm 0.76$ |
| Llama-3.1-70B-Instruct | 67.7 | $66.51 \pm 0.15$ | $63.79 \pm 1.21$ | $58.41 \pm 0.57$ | $49.14 \pm 1$ | $30.58 \pm 0.72$ |
| Qwen2.5-7B-Instruct-1M | 55.56 | $50.7 \pm 1.3$ | $45.73 \pm 0.66$ | $39.22 \pm 0.17$ | $36.85 \pm 0.38$ | $33.66 \pm 0.47$ |
| Qwen2.5-14B-Instruct-1M | 61.76 | $58.32 \pm 0.73$ | $53.95 \pm 1.4$ | $48.86 \pm 0.75$ | $44.16 \pm 0.7$ | $37.7 \pm 0.42$ |
| Gemma-3-12B-IT | 67.49 | $62.66 \pm 0.97$ | $57.18 \pm 0.78$ | $52.39 \pm 1.69$ | $45.88 \pm 1.31$ | $33.39 \pm 0.42$ |
| Gemma-3-27B-IT | 67.71 | $63.17 \pm 0.65$ | $60.53 \pm 0.26$ | $53.19 \pm 1.44$ | $47.35 \pm 0.94$ | $38.93 \pm 1.21$ |
| GPT-4.1 mini | 70.19 | $69.04 \pm 0.27$ | $67.41 \pm 0.26$ | $64.57 \pm 0.86$ | $60.28 \pm 0.4$ | $56.84 \pm 0.36$ |
| Qwen3-8B | 71.66 | $71.3 \pm 0.81$ | $71.02 \pm 0.82$ | $67.13 \pm 0.39$ | $58.53 \pm 1.33$ | $53.48 \pm 1.55$ |
| Qwen3-14B | 71.9 | $71.39 \pm 0.43$ | $69.8 \pm 1.18$ | $68.39 \pm 0.69$ | $66.22 \pm 0.41$ | $61.1 \pm 0.62$ |
| Qwen3-32B | 71.32 | $71.73 \pm 0.46$ | $71.22 \pm 0.9$ | $68.94 \pm 1.1$ | $67.71 \pm 0.61$ | $62.11 \pm 1.81$ |
| Gemini 2.5 Flash-Lite | 71.6 | $72.16 \pm 0.89$ | $72.07 \pm 0.73$ | $69.52 \pm 1.35$ | $67.66 \pm 0.19$ | $65.61 \pm 0.34$ |
| o4-mini | 75.95 | $75.6 \pm 0.41$ | $74.73 \pm 0.43$ | $73.39 \pm 0.2$ | $70.48 \pm 1.09$ | $67.24 \pm 0.57$ |

Table 13: Static NIAH performance in F1 score ($\times 10^{-2}, \uparrow$) using Qwen3-0.6B for haystack construction, where we average the results over three random Haystack orderings. 0 stands for the case without distractors.

| Context Size (# Tokens) | 0 | 8K | 16K | 32K | 64K | 128K |
|---|---|---|---|---|---|---|
| Llama-3.1-8B-Instruct | 59.8 | $54.87 \pm 0.75$ | $50.71 \pm 0.89$ | $43.41 \pm 0.6$ | $39.08 \pm 1.00$ | $33.33 \pm 0.5$ |
| Llama-3.1-70B-Instruct | 67.7 | $65.69 \pm 0.77$ | $62.52 \pm 1.01$ | $56.81 \pm 1.1$ | $47.87 \pm 0.2$ | $26.49 \pm 0.73$ |
| Qwen2.5-7B-Instruct-1M | 55.56 | $50.79 \pm 1.29$ | $46.75 \pm 0.85$ | $41.05 \pm 1.00$ | $34.85 \pm 0.50$ | $29.55 \pm 0.46$ |
| Qwen2.5-14B-Instruct-1M | 61.76 | $57.1 \pm 0.39$ | $52.83 \pm 1.81$ | $48.7 \pm 0.22$ | $40.78 \pm 0.88$ | $35.36 \pm 0.28$ |
| Gemma-3-12B-IT | 67.49 | $61.2 \pm 0.2$ | $55.51 \pm 1.0$ | $51.24 \pm 0.46$ | $42.66 \pm 0.79$ | $32.37 \pm 1.66$ |
| Gemma-3-27B-IT | 67.71 | $62.87 \pm 0.39$ | $57.77 \pm 0.26$ | $51.71 \pm 0.37$ | $42.95 \pm 1.05$ | $34.67 \pm 0.38$ |
| GPT-4.1 mini | 70.19 | $68.2 \pm 0.53$ | $66.09 \pm 1.67$ | $63.57 \pm 0.45$ | $59.97 \pm 0.55$ | $56.74 \pm 1.38$ |
| Qwen3-8B | 71.66 | $71.69 \pm 0.81$ | $69.9 \pm 0.52$ | $64.12 \pm 0.82$ | $54.24 \pm 0.41$ | $48.89 \pm 1.62$ |
| Qwen3-14B | 71.9 | $69.83 \pm 0.98$ | $69.04 \pm 1.14$ | $68.03 \pm 0.13$ | $64.21 \pm 0.52$ | $56.77 \pm 1.45$ |
| Qwen3-32B | 71.32 | $70.66 \pm 1.22$ | $70.29 \pm 0.76$ | $68.68 \pm 0.56$ | $64.65 \pm 0.39$ | $57.11 \pm 1.22$ |
| Gemini 2.5 Flash-Lite | 71.6 | $71.39 \pm 0.49$ | $70.61 \pm 0.68$ | $69.67 \pm 0.87$ | $67.39 \pm 0.35$ | $62.58 \pm 0.64$ |
| o4-mini | 75.95 | $74.54 \pm 0.17$ | $75.33 \pm 0.43$ | $72.93 \pm 0.44$ | $69.72 \pm 0.59$ | $63.91 \pm 1.39$ |

Table 14: Static NIAH performance in F1 score ($\times 10^{-2}, \uparrow$) using hybrid retriever for haystack construction, where we average the results over three random Haystack orderings. 0 stands for the case without distractors.

| Context Size (# Tokens) | 0 | 8K | 16K | 32K | 64K | 128K |
|---|---|---|---|---|---|---|
| Llama-3.1-8B-Instruct | 59.8 | $54.95 \pm 1.16$ | $51.33 \pm 0.85$ | $43.77 \pm 1.42$ | $37.42 \pm 0.92$ | $33.34 \pm 0.58$ |
| Llama-3.1-70B-Instruct | 67.7 | $66.29 \pm 0.19$ | $62.54 \pm 0.25$ | $57.51 \pm 0.05$ | $47.76 \pm 1.77$ | $28.82 \pm 1.01$ |
| Qwen2.5-7B-Instruct-1M | 55.56 | $52.46 \pm 1.21$ | $46.7 \pm 1.14$ | $40.32 \pm 0.78$ | $35.26 \pm 1.37$ | $32.52 \pm 0.61$ |
| Qwen2.5-14B-Instruct-1M | 61.76 | $58.28 \pm 0.29$ | $52.69 \pm 0.46$ | $47.98 \pm 0.8$ | $42.9 \pm 0.71$ | $36.16 \pm 1.04$ |
| Gemma-3-12B-IT | 67.49 | $62.41 \pm 0.43$ | $56.83 \pm 0.77$ | $51.66 \pm 0.25$ | $44.11 \pm 0.63$ | $32.51 \pm 1.72$ |
| Gemma-3-27B-IT | 67.71 | $64.26 \pm 0.43$ | $57.56 \pm 0.35$ | $52.08 \pm 0.22$ | $44.95 \pm 1.07$ | $36.85 \pm 1.16$ |
| GPT-4.1 mini | 70.19 | $68.46 \pm 0.53$ | $66.4 \pm 0.81$ | $62.48 \pm 0.39$ | $59.72 \pm 0.38$ | $56.77 \pm 0.81$ |
| Qwen3-8B | 71.66 | $72.03 \pm 0.68$ | $69.57 \pm 0.42$ | $65.96 \pm 0.97$ | $57.68 \pm 0.65$ | $51.85 \pm 0.86$ |
| Qwen3-14B | 71.9 | $70.21 \pm 0.15$ | $69.97 \pm 0.4$ | $66.74 \pm 0.37$ | $65.4 \pm 0.71$ | $59.58 \pm 1.13$ |
| Qwen3-32B | 71.32 | $71.61 \pm 0.69$ | $70.66 \pm 0.84$ | $68.19 \pm 1.14$ | $63.68 \pm 0.59$ | $57.99 \pm 0.65$ |
| Gemini 2.5 Flash-Lite | 71.6 | $71.56 \pm 0.81$ | $70.04 \pm 0.6$ | $69.24 \pm 0.78$ | $66.75 \pm 0.65$ | $64.06 \pm 0.51$ |
| o4-mini | 75.95 | $74.67 \pm 0.32$ | $74.73 \pm 1.23$ | $72.88 \pm 0.38$ | $70.22 \pm 0.18$ | $65.64 \pm 0.2$ |

Table 15: Static NIAH performance in F1 score ($\times 10^{-2}, \uparrow$) using BM25 + PPR for haystack construction, where we average the results over three random Haystack orderings. 0 stands for the case without distractors.

| Context Size (# Tokens) | 0 | 8K | 16K | 32K | 64K | 128K |
|---|---|---|---|---|---|---|
| Llama-3.1-8B-Instruct | 59.8 | $56.76 \pm 0.34$ | $51.34 \pm 0.64$ | $45.13 \pm 1.19$ | $41.23 \pm 1.16$ | $37.44 \pm 2.2$ |
| Llama-3.1-70B-Instruct | 67.7 | $66.83 \pm 0.41$ | $63.39 \pm 0.96$ | $59.46 \pm 0.95$ | $50.4 \pm 1.22$ | $33.53 \pm 1.22$ |
| Qwen2.5-7B-Instruct-1M | 55.56 | $52.14 \pm 0.51$ | $45.37 \pm 1.58$ | $42.73 \pm 1.63$ | $37.8 \pm 1.88$ | $37.74 \pm 0.19$ |
| Qwen2.5-14B-Instruct-1M | 61.76 | $58.73 \pm 0.73$ | $53.72 \pm 1.03$ | $49.81 \pm 1.26$ | $44.38 \pm 1.29$ | $41.94 \pm 0.79$ |
| Gemma-3-12B-IT | 67.49 | $62.8 \pm 0.42$ | $58.08 \pm 0.49$ | $52.37 \pm 0.85$ | $45.95 \pm 1.31$ | $38.5 \pm 0.21$ |
| Gemma-3-27B-IT | 67.71 | $64.31 \pm 0.83$ | $60.06 \pm 0.64$ | $53.15 \pm 1.13$ | $49.49 \pm 0.64$ | $42.65 \pm 0.94$ |
| GPT-4.1 mini | 70.19 | $69.12 \pm 0.28$ | $67.36 \pm 0.96$ | $64.48 \pm 0.71$ | $60.93 \pm 0.7$ | $59.26 \pm 1.29$ |
| Qwen3-8B | 71.66 | $71.71 \pm 0.35$ | $70.6 \pm 0.78$ | $66.74 \pm 0.81$ | $60.06 \pm 0.4$ | $57.87 \pm 0.37$ |
| Qwen3-14B | 71.9 | $70.93 \pm 0.11$ | $70.24 \pm 0.81$ | $68.22 \pm 0.51$ | $66.72 \pm 1.13$ | $62.8 \pm 0.65$ |
| Qwen3-32B | 71.32 | $71.48 \pm 0.25$ | $70.86 \pm 0.26$ | $69.2 \pm 0.96$ | $66.73 \pm 0.96$ | $63.91 \pm 0.68$ |
| Gemini 2.5 Flash-Lite | 71.6 | $72.66 \pm 1.48$ | $72.69 \pm 0.66$ | $69.43 \pm 0.96$ | $66.61 \pm 0.52$ | $65.64 \pm 0.26$ |
| o4-mini | 75.95 | $75.86 \pm 0.71$ | $74.77 \pm 0.8$ | $73.73 \pm 0.22$ | $71.73 \pm 0.36$ | $68.96 \pm 0.47$ |

Table 16: Static NIAH performance in F1 score $(\times 10^{-2}, \uparrow)$ using Qwen3-0.6B + PPR for haystack construction, where we average the results over three random Haystack orderings. 0 stands for the case without distractors.

| Context Size (# Tokens) | 0 | 8K | 16K | 32K | 64K | 128K |
|---|---|---|---|---|---|---|
| Llama-3.1-8B-Instruct | 59.8 | $55.22 \pm 1.19$ | $51.4 \pm 1.12$ | $46.64 \pm 0.94$ | $41.6 \pm 1.04$ | $36.71 \pm 0.6$ |
| Llama-3.1-70B-Instruct | 67.7 | $66.49 \pm 0.78$ | $62.85 \pm 0.07$ | $57.63 \pm 0.3$ | $50.4 \pm 0.95$ | $33.35 \pm 0.65$ |
| Qwen2.5-7B-Instruct-1M | 55.56 | $51.54 \pm 0.68$ | $47.34 \pm 1.24$ | $44.77 \pm 0.48$ | $41.14 \pm 1.06$ | $36.05 \pm 1.55$ |
| Qwen2.5-14B-Instruct-1M | 61.76 | $58.45 \pm 0.29$ | $53.54 \pm 0.49$ | $49.4 \pm 1.65$ | $44.76 \pm 1.43$ | $39.46 \pm 1.15$ |
| Gemma-3-12B-IT | 67.49 | $62.78 \pm 0.38$ | $56.4 \pm 0.75$ | $53.29 \pm 0.87$ | $46.8 \pm 0.77$ | $38.47 \pm 0.21$ |
| Gemma-3-27B-IT | 67.71 | $63.72 \pm 0.52$ | $59.01 \pm 1.53$ | $54.35 \pm 0.84$ | $49.01 \pm 1.02$ | $40.79 \pm 0.93$ |
| GPT-4.1 mini | 70.19 | $68.08 \pm 1.18$ | $66.18 \pm 0.3$ | $63.23 \pm 0.56$ | $61.09 \pm 0.61$ | $57.49 \pm 0.31$ |
| Qwen3-8B | 71.66 | $71.53 \pm 0.59$ | $69.82 \pm 0.89$ | $67.09 \pm 0.52$ | $58.84 \pm 1.45$ | $57.84 \pm 1.53$ |
| Qwen3-14B | 71.9 | $70.46 \pm 1.0$ | $70.19 \pm 0.77$ | $67.61 \pm 0.1$ | $66.08 \pm 0.66$ | $61.4 \pm 0.64$ |
| Qwen3-32B | 71.32 | $70.63 \pm 0.57$ | $70.19 \pm 0.22$ | $68.77 \pm 1.35$ | $65.64 \pm 0.47$ | $61.54 \pm 1.12$ |
| Gemini 2.5 Flash-Lite | 71.6 | $72.13 \pm 1.34$ | $70.21 \pm 0.25$ | $69.26 \pm 0.77$ | $67.64 \pm 0.64$ | $65.26 \pm 1.6$ |
| o4-mini | 75.95 | $74.49 \pm 0.38$ | $74.35 \pm 0.19$ | $73.86 \pm 0.51$ | $70.81 \pm 0.63$ | $67.71 \pm 0.41$ |

Table 17: Static NIAH performance in F1 score $(\times 10^{-2}, \uparrow)$ using hybrid + PPR retriever for haystack construction, where we average the results over three random Haystack orderings. 0 stands for the case without distractors.

| Context Size (# Tokens) | 0 | 8K | 16K | 32K | 64K | 128K |
|---|---|---|---|---|---|---|
| Llama-3.1-8B-Instruct | 59.8 | $55.19 \pm 1.37$ | $51.69 \pm 0.71$ | $45.57 \pm 1.02$ | $41.97 \pm 0.54$ | $38.99 \pm 1.78$ |
| Llama-3.1-70B-Instruct | 67.7 | $66.97 \pm 0.78$ | $62.81 \pm 1.17$ | $58.75 \pm 0.6$ | $51.36 \pm 0.61$ | $33.16 \pm 1.12$ |
| Qwen2.5-7B-Instruct-1M | 55.56 | $52.21 \pm 1.02$ | $47.8 \pm 0.19$ | $44.18 \pm 0.38$ | $40.59 \pm 0.49$ | $39.17 \pm 1.27$ |
| Qwen2.5-14B-Instruct-1M | 61.76 | $58.03 \pm 0.55$ | $53.63 \pm 0.79$ | $50.8 \pm 0.69$ | $46.9 \pm 1.25$ | $42.15 \pm 0.7$ |
| Gemma-3-12B-IT | 67.49 | $62.2 \pm 0.98$ | $57.27 \pm 1.04$ | $53.32 \pm 0.47$ | $48.2 \pm 0.88$ | $37.51 \pm 4.94$ |
| Gemma-3-27B-IT | 67.71 | $64.02 \pm 1.13$ | $58.84 \pm 0.49$ | $55.6 \pm 0.68$ | $50.27 \pm 0.14$ | $41.95 \pm 1.08$ |
| GPT-4.1 mini | 70.19 | $68.7 \pm 0.76$ | $66.51 \pm 0.81$ | $64.43 \pm 0.61$ | $61.79 \pm 1.28$ | $60.75 \pm 0.31$ |
| Qwen3-8B | 71.66 | $71.5 \pm 0.71$ | $69.7 \pm 0.32$ | $66.95 \pm 0.51$ | $60.46 \pm 0.28$ | $58.62 \pm 0.52$ |
| Qwen3-14B | 71.9 | $70.14 \pm 0.32$ | $68.83 \pm 0.74$ | $66.87 \pm 2.12$ | $66.81 \pm 0.78$ | $62.13 \pm 0.47$ |
| Qwen3-32B | 71.32 | $71.96 \pm 0.67$ | $71.03 \pm 0.57$ | $70.13 \pm 0.43$ | $65.37 \pm 0.55$ | $63.4 \pm 0.62$ |
| Gemini 2.5 Flash-Lite | 71.6 | $71.57 \pm 0.87$ | $70.39 \pm 0.6$ | $69.07 \pm 0.35$ | $66.42 \pm 0.4$ | $65.27 \pm 0.52$ |
| o4-mini | 75.95 | $75.59 \pm 0.19$ | $74.14 \pm 0.7$ | $73.68 \pm 0.23$ | $71.03 \pm 0.61$ | $68.13 \pm 0.4$ |

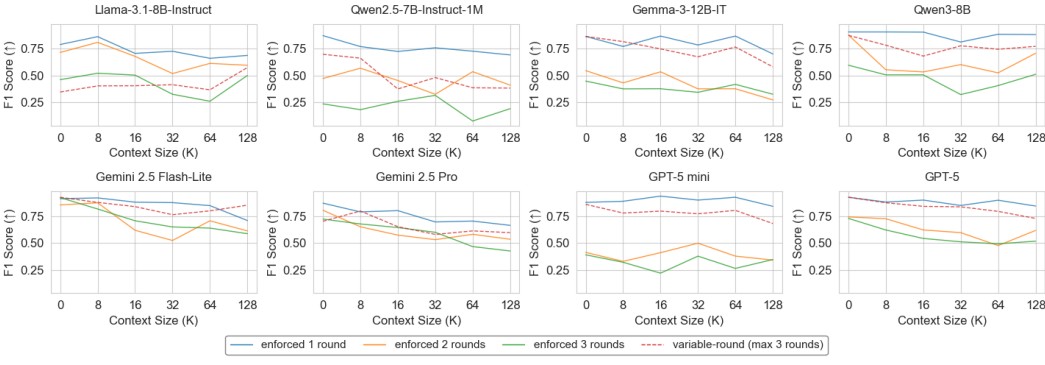

Figure 9: Dynamic NIAH performance for single-hop questions, where the haystack is constructed with BM25 + PPR. 0 stands for the case without retrieval-based distractors.

Table 18: Dynamic NIAH performance in F1 score ($\times 10^{-2}, \uparrow$) using BM25 + PPR for haystack construction, where retriever-ranked haystack ordering is used. 0 stands for the case without distractors.

| Context Size (# Tokens) | # Rounds | 0 | 8K | 16K | 32K | 64K | 128K |
|---|---|---|---|---|---|---|---|
| Llama-3.1-8B-Instruct | 1 | 56.86 | 60.46 | 52.06 | 51.93 | 42.42 | 39.97 |
| | 2 | 50.88 | 49.52 | 51.41 | 39.18 | 37.71 | 27.88 |
| | 3 | 26.51 | 33.79 | 37.04 | 28.63 | 23.79 | 32.77 |
| | max 3 | 34.52 | 43.16 | 34.85 | 31.62 | 30.45 | 29.66 |
| Qwen2.5-7B-Instruct-1M | 1 | 60.59 | 55.16 | 52.63 | 45.96 | 46.3 | 48.12 |
| | 2 | 37.64 | 46.31 | 37.53 | 28.57 | 34.78 | 36.8 |
| | 3 | 24 | 25.35 | 28.14 | 22.88 | 20.98 | 21.93 |
| | max 3 | 43.61 | 38.99 | 35.95 | 36.51 | 34.5 | 34.97 |
| Gemma-3-12B-IT | 1 | 66.54 | 63.48 | 61.94 | 57.27 | 54.64 | 45.51 |
| | 2 | 48.91 | 47.96 | 53.82 | 40.63 | 41.77 | 36.3 |
| | 3 | 39.07 | 38.94 | 40.31 | 32.48 | 35.18 | 29.28 |
| | max 3 | 64.88 | 60.1 | 62.19 | 52.01 | 51.11 | 40.87 |
| Qwen3-8B | 1 | 71.98 | 74.3 | 74.28 | 71.93 | 67.54 | 68.72 |
| | 2 | 62.89 | 58.54 | 58.24 | 53.1 | 54.79 | 54.58 |
| | 3 | 49.87 | 47.34 | 52.3 | 42.98 | 46.79 | 46.22 |
| | max 3 | 69.06 | 69.76 | 63.86 | 64.25 | 59.01 | 59.83 |
| Gemini 2.5 Flash-Lite | 1 | 74.62 | 73.29 | 72.67 | 72.42 | 69.4 | 65.38 |
| | 2 | 69.3 | 72.75 | 64.79 | 61.44 | 63.12 | 61.78 |
| | 3 | 71.46 | 71.25 | 63.92 | 63.2 | 60.49 | 57.03 |
| | max 3 | 71.55 | 70.97 | 69.73 | 65.83 | 65.97 | 64.98 |
| Gemini 2.5 Pro | 1 | 75.86 | 72.78 | 74.97 | 70.45 | 69.34 | 69.65 |
| | 2 | 72.66 | 65.99 | 64.39 | 60.68 | 60.98 | 60.88 |
| | 3 | 69.06 | 70.38 | 66.36 | 58.63 | 55.98 | 54.75 |
| | max 3 | 68.36 | 71.08 | 67.85 | 63.87 | 66.35 | 62.03 |
| GPT-5 mini | 1 | 79.87 | 78.04 | 79.1 | 75.78 | 75.27 | 72.51 |
| | 2 | 56.48 | 58.96 | 58.24 | 55.54 | 52.15 | 51.79 |
| | 3 | 40.81 | 41 | 32.89 | 38.66 | 36.39 | 40.74 |
| | max 3 | 73.47 | 70.4 | 71.42 | 68.2 | 71.81 | 65.46 |
| GPT-5 | 1 | 78.28 | 77.36 | 77.2 | 75.03 | 72.98 | 75.15 |
| | 2 | 72.72 | 72.33 | 68.73 | 70.17 | 66.29 | 68.58 |
| | 3 | 71.7 | 70.13 | 67.2 | 66.45 | 64.07 | 63.31 |
| | max 3 | 76.47 | 74.89 | 75.06 | 73.56 | 74.03 | 71.96 |

Table 19: Dynamic NIAH performance in F1 score ($\times 10^{-2}, \uparrow$) using Qwen3-0.6B for haystack construction, where retriever-ranked haystack ordering is used. 0 stands for the case without distractors.

| Context Size (# Tokens) | # Rounds | 0 | 8K | 16K | 32K | 64K | 128K |
|---|---|---|---|---|---|---|---|
| Llama-3.1-8B-Instruct | 1 | 56.86 | 56.76 | 53.41 | 49.45 | 42.81 | 30.15 |
| | 2 | 56.68 | 45.59 | 51.41 | 39.18 | 37.71 | 27.88 |
| | 3 | 36.68 | 30.48 | 38.74 | 27.87 | 29.65 | 26.08 |
| | max 3 | 38.5 | 31.12 | 37.8 | 31.7 | 33.55 | 25.24 |
| Qwen2.5-7B-Instruct-1M | 1 | 60.59 | 55.45 | 50.54 | 44.36 | 39.97 | 35.51 |
| | 2 | 32.44 | 35.22 | 37.24 | 37.41 | 38.49 | 35.63 |
| | 3 | 26.1 | 23.45 | 22.02 | 24.84 | 20.33 | 27.18 |
| | max 3 | 44.36 | 36.92 | 33.9 | 31.46 | 38.7 | 38.12 |
| Gemma-3-12B-IT | 1 | 66.54 | 63.77 | 58.67 | 52.66 | 55.18 | 47.59 |
| | 2 | 55.13 | 51.63 | 49.39 | 47.28 | 44.81 | 35.55 |
| | 3 | 40.07 | 43.32 | 37.54 | 32.23 | 32.57 | 23.7 |
| | max 3 | 61.37 | 63.83 | 59.1 | 58.86 | 57.05 | 50.16 |
| Qwen3-8B | 1 | 71.98 | 69.61 | 65.56 | 69.44 | 62.41 | 55.57 |
| | 2 | 58.95 | 59.54 | 62.48 | 63.91 | 57.94 | 49.83 |
| | 3 | 49.26 | 46.95 | 50.32 | 49.84 | 40.14 | 41.64 |
| | max 3 | 69.48 | 65.31 | 63.26 | 59.78 | 55.2 | 54.32 |
| Gemini 2.5 Flash-Lite | 1 | 74.62 | 72.25 | 73.99 | 71.92 | 70.23 | 63.28 |
| | 2 | 68.04 | 66.85 | 65.56 | 64.87 | 62.67 | 62.76 |
| | 3 | 69.62 | 64.97 | 64.6 | 58.91 | 63.56 | 63.23 |
| | max 3 | 69.35 | 66.77 | 67.34 | 68.09 | 63.23 | 64.87 |
| Gemini 2.5 Pro | 1 | 75.86 | 73.34 | 72.20 | 73.9 | 69.61 | 68.52 |
| | 2 | 69.07 | 65.81 | 65.95 | 62.05 | 61.16 | 58.08 |
| | 3 | 67.26 | 63.84 | 62.91 | 61.02 | 53.09 | 51.43 |
| | max 3 | 71.37 | 68.28 | 65.16 | 63.12 | 59.73 | 60.93 |
| GPT-5 mini | 1 | 79.87 | 77.18 | 76.97 | 73.68 | 72.14 | 70.14 |
| | 2 | 60.24 | 57.4 | 57.2 | 53.82 | 51.23 | 53.33 |
| | 3 | 39.63 | 41.02 | 38.99 | 35.26 | 37.73 | 37.92 |
| | max 3 | 73.27 | 71.7 | 70.77 | 68.98 | 67.59 | 65.59 |
| GPT-5 | 1 | 78.28 | 74.8 | 76.31 | 76.98 | 73.47 | 71.7 |
| | 2 | 71.92 | 69.38 | 70.28 | 72.96 | 68.26 | 66.3 |
| | 3 | 71.21 | 68.35 | 63.01 | 68.52 | 63.35 | 65.48 |
| | max 3 | 76.22 | 76.16 | 71.92 | 71.15 | 72.67 | 72.95 |

Table 20: % cases where early stop happens in variable-round dynamic NIAH. BM25 + PPR is used for haystack construction, and retriever-ranked haystack ordering is used. 0 stands for the case without distractors.

| Context Size (# Tokens) | 0 | 8K | 16K | 32K | 64K | 128K |
|---|---|---|---|---|---|---|
| Llama-3.1-8B-Instruct | 13% | 16% | 7% | 15% | 11% | 12% |
| Qwen2.5-7B-Instruct-1M | 55% | 57% | 52% | 44% | 28% | 24% |
| Gemma-3-12B-IT | 91% | 89% | 87% | 85% | 85% | 91% |
| Qwen3-8B | 97% | 99% | 100% | 99% | 96% | 97% |
| Gemini 2.5 Flash-Lite | 95% | 97% | 97% | 94% | 98% | 94% |
| Gemini 2.5 Pro | 92% | 94% | 92% | 90% | 87% | 86% |
| GPT-5 mini | 97% | 97% | 98% | 96% | 97% | 94% |
| GPT-5 | 99% | 97% | 97% | 97% | 97% | 99% |

Table 21: % cases where early stop happens in variable-round dynamic NIAH. Qwen3-0.6B is used for haystack construction, and retriever-ranked haystack ordering is used. 0 stands for the case without distractors.

| Context Size (# Tokens) | 0 | 8K | 16K | 32K | 64K | 128K |
|---|---|---|---|---|---|---|
| Llama-3.1-8B-Instruct | 22% | 16% | 16% | 12% | 16% | 7% |
| Qwen2.5-7B-Instruct-1M | 59% | 52% | 47% | 38% | 28% | 21% |
| Gemma-3-12B-IT | 88% | 92% | 86% | 90% | 89% | 91% |
| Qwen3-8B | 97% | 99% | 98% | 98% | 96% | 95% |
| Gemini 2.5 Flash-Lite | 94% | 93% | 93% | 94% | 94% | 95% |
| Gemini 2.5 Pro | 95% | 89% | 90% | 88% | 84% | 86% |
| GPT-5 mini | 97% | 94% | 95% | 93% | 97% | 93% |
| GPT-5 | 98% | 97% | 98% | 98% | 98% | 99% |

Table 22: Dynamic NIAH performance in F1 score ($\times 10^{-2}, \uparrow$) for single-hop questions. BM25 + PPR is used for haystack construction, and retriever-ranked haystack ordering is used. 0 stands for the case without distractors.

| Context Size (# Tokens) | # Rounds | 0 | 8K | 16K | 32K | 64K | 128K |
|---|---|---|---|---|---|---|---|
| Llama-3.1-8B-Instruct | 1 | 78.98 | 86.3 | 70.69 | 72.67 | 66.11 | 68.8 |
| | 2 | 71.6 | 80.97 | 67.92 | 51.88 | 61.36 | 59.6 |
| | 3 | 46.19 | 52.18 | 50.42 | 32.56 | 26.04 | 50.07 |
| | max 3 | 34.65 | 40.37 | 40.49 | 41.48 | 36.73 | 57.45 |
| Qwen2.5-7B-Instruct-1M | 1 | 87.25 | 77.06 | 72.5 | 75.88 | 72.74 | 69.26 |
| | 2 | 47.27 | 56.85 | 45.4 | 32.53 | 53.62 | 41.11 |
| | 3 | 23.6 | 18.2 | 26 | 31.53 | 7.62 | 19.11 |
| | max 3 | 69.84 | 66.25 | 37.51 | 48.13 | 38.62 | 38.35 |
| Gemma-3-12B-IT | 1 | 86.54 | 77.16 | 86.85 | 78.52 | 86.85 | 70.19 |
| | 2 | 54.63 | 43.19 | 53.5 | 37.6 | 37.69 | 27.29 |
| | 3 | 44.87 | 37.58 | 37.71 | 34.38 | 41.79 | 32.6 |
| | max 3 | 86.54 | 81.67 | 74.88 | 67.52 | 76.64 | 58.15 |
| Qwen3-8B | 1 | 90.69 | 90.69 | 90.56 | 81.3 | 88.54 | 88.24 |
| | 2 | 87.84 | 55.42 | 53.38 | 60.23 | 52.53 | 70.77 |
| | 3 | 59.66 | 50.67 | 50.68 | 32.28 | 40.66 | 51.16 |
| | max 3 | 87.49 | 78.33 | 68.19 | 77.76 | 74.55 | 77.25 |
| Gemini 2.5 Flash-Lite | 1 | 91.48 | 92.19 | 88.24 | 87.78 | 85 | 71.11 |
| | 2 | 85.64 | 87.62 | 61.98 | 52.5 | 70.83 | 61.41 |
| | 3 | 92.45 | 81.86 | 70.87 | 65.12 | 64.03 | 58.81 |
| | max 3 | 92.8 | 87.99 | 84.07 | 76.53 | 80.13 | 85.4 |
| Gemini 2.5 Pro | 1 | 87.31 | 79.13 | 80.32 | 69.79 | 70.58 | 66.59 |
| | 2 | 80.67 | 65.25 | 57.49 | 53.24 | 58.24 | 53.61 |
| | 3 | 72.45 | 67.9 | 64.54 | 60.11 | 46.79 | 42.68 |
| | max 3 | 70.43 | 79.97 | 65.38 | 58.12 | 61.43 | 59.78 |
| GPT-5 mini | 1 | 88.01 | 89.05 | 93.86 | 90.16 | 92.8 | 84.34 |
| | 2 | 41.4 | 33 | 41.13 | 49.94 | 37.89 | 34.21 |
| | 3 | 39.12 | 32.14 | 22.01 | 37.88 | 26.42 | 34.71 |
| | max 3 | 86.21 | 78.06 | 79.86 | 77.38 | 80.54 | 68.31 |
| GPT-5 | 1 | 92.8 | 88.31 | 90.16 | 85.13 | 90.03 | 84.6 |
| | 2 | 74.5 | 72.75 | 62.31 | 59.79 | 47.73 | 61.84 |
| | 3 | 73.07 | 62.26 | 54.38 | 51.24 | 49.37 | 51.85 |
| | max 3 | 92.8 | 87.66 | 84.33 | 83.86 | 79.63 | 73.05 |

Table 23: Dynamic NIAH performance (ablation setting) in F1 score ($\times 10^{-2}$, $\uparrow$) using Qwen3-0.6B for haystack construction, where retriever-ranked haystack ordering is used. 0 stands for the case without distractors.

| Context Size (# Tokens) | # Rounds | 0 | 8K | 16K | 32K | 64K | 128K |
|---|---|---|---|---|---|---|---|
| Llama-3.1-8B-Instruct | 1 | 56.86 | 56.76 | 53.41 | 49.45 | 42.81 | 30.15 |
| | 2 | 57.08 | 53.00 | 51.53 | 53.79 | 52.28 | 43.2 |
| | 3 | 53.69 | 53.45 | 51.82 | 46.9 | 50.74 | 42.39 |
| | max 3 | 4.54 | 3.1 | 3.19 | 1.81 | 1.78 | 1.76 |
| Qwen2.5-7B-Instruct-1M | 1 | 60.59 | 55.45 | 50.54 | 44.36 | 39.97 | 35.51 |
| | 2 | 57.02 | 53.88 | 57.24 | 51.94 | 50.56 | 52.84 |
| | 3 | 53.6 | 52.67 | 53.22 | 56.13 | 52.24 | 52.85 |
| | max 3 | 20.37 | 22.41 | 24.12 | 22.37 | 19.03 | 18.57 |
| Gemma-3-12B-IT | 1 | 66.54 | 63.77 | 58.67 | 52.66 | 55.18 | 47.59 |
| | 2 | 54.21 | 55.36 | 52.66 | 49.34 | 49.87 | 50.38 |
| | 3 | 58.31 | 57.44 | 55.89 | 51.42 | 53.52 | 46.91 |
| | max 3 | 57.21 | 53.38 | 49.93 | 36.97 | 40.04 | 28.68 |
| Qwen3-8B | 1 | 71.98 | 69.61 | 65.56 | 69.44 | 62.41 | 55.57 |
| | 2 | 68.4 | 66.33 | 64.09 | 58.26 | 59.10 | 51.26 |
| | 3 | 65.07 | 64.06 | 66.56 | 57.9 | 56.87 | 57.79 |
| | max 3 | 67.18 | 62.89 | 63.04 | 58.96 | 55.78 | 55.28 |
| Gemini 2.5 Flash-Lite | 1 | 74.62 | 72.25 | 73.99 | 71.92 | 70.23 | 63.28 |
| | 2 | 70.17 | 67.17 | 64.80 | 64.01 | 65.04 | 65.48 |
| | 3 | 65.68 | 69.59 | 67.58 | 62.81 | 61.25 | 61.02 |
| | max 3 | 67.94 | 61.87 | 64.77 | 61.79 | 58.26 | 58.43 |
| Gemini 2.5 Pro | 1 | 75.86 | 73.34 | 72.20 | 73.9 | 69.61 | 68.52 |
| | 2 | 58.4 | 57.57 | 59.57 | 53.34 | 50.46 | 50.79 |
| | 3 | 58.49 | 53.85 | 54.66 | 49.65 | 46.12 | 47.33 |
| | max 3 | 65.93 | 62.9 | 60.99 | 55.81 | 56.44 | 54.36 |
| GPT-5 mini | 1 | 79.87 | 77.18 | 76.97 | 73.68 | 72.14 | 70.14 |
| | 2 | 64.88 | 63.14 | 64.02 | 61.25 | 63.62 | 61.51 |
| | 3 | 65.73 | 63.07 | 61.84 | 60.76 | 60.08 | 58.86 |
| | max 3 | 69.29 | 67.56 | 64.99 | 66.05 | 65.9 | 69.65 |
| GPT-5 | 1 | 78.28 | 74.8 | 76.31 | 76.98 | 73.47 | 71.7 |
| | 2 | 74.29 | 70.93 | 72.36 | 69.77 | 69.03 | 71.65 |
| | 3 | 73.53 | 69.21 | 71.29 | 68.86 | 67.43 | 70.11 |
| | max 3 | 71.69 | 67.79 | 68.67 | 69.66 | 66.04 | 71.06 |

