# OpenReview forum: "Haystack Engineering: Context Engineering for Heterogeneous and Agentic Long-Context Evaluation"
_ICLR.cc/2026/Conference — Submitted to ICLR 2026_

### Official Review · Reviewer_L4UT · 2025-10-24

**Soundness:** 3
**Presentation:** 1
**Contribution:** 2
**Rating:** 2
**Confidence:** 3

**Summary:**

This paper introduces "haystack engineering" to create more realistic benchmarks for long-context LLMs, arguing that synthetic "Needle-in-a-Haystack" (NIAH) tests fail to capture real-world noise. The authors present HaystackCraft, a new benchmark built on Wikipedia, which evaluates models in two settings: (1) Static NIAH, testing how performance is affected by distractors from heterogeneous retrieval strategies (sparse, dense, graph-based), and (2) Dynamic NIAH, a novel agentic setting that tests robustness to cascading errors from multi-step query refinement. Key findings show that graph-based (PPR) reranking can mitigate harmful distractors in the static setting. More importantly, in the dynamic setting, even SOTA models like Gemini 2.5 Pro and GPT-5 are brittle, with performance degrading over multiple reasoning rounds, demonstrating that models are currently more robust to "width" (long contexts) than "depth" (reasoning iterations).

**Strengths:**

1. This paper develops more realistic evaluations (dynamic, LLM-dependent, agentic) for long-context models to tackle the limitations of synthetic NIAH benchmarks. The "haystack engineering" concept is a clear and valuable contribution to this effort.
2. This paper has a thorough experiments among the retrieval strategies under static NIAH settings.

**Weaknesses:**

1. The figures in this work is confusing and didn't show the main point. In Figure3, the authors want to compare different retrieval strategy then it is better to show one figure with different strategies and the scores are average on all 12 models.  In Figure4, the authors compare retriever-ranked and random haystack ordering, but there is no comparison in the figure and we need to compare Figure3 and Figure4 to understand.
2. The paper complicated its simple idea with many unnecessary explanations. For example, the idea of using retrieved documents as haystack is already well-used by most of works, but this paper used section 3.1, 3.2, 3.4 to explain this. On the other hand, the paper should discuss novel dynamic NIAH in details. The paper should be well written with concise data/task introduction and detailed analysis without exceeding 9 page limit.

**Questions:**

1. In Figure2, the authors claim reranking with PRR consistently boosts performance, especially for multi-hop questions. However, 4-hop shows opposite results. Can you explain why?
2. Can you show some examples that how graph-based reranking mitigates harmful distractors?

---

> ### Author Response · Authors · 2025-11-27
>
> Thank you for your detailed feedback. Below, we address your questions and concerns.
>
> **Q1**: Figure 3 is confusing and does not show the main point. As the authors want to compare different retrieval strategies, it is better to show one figure where the scores for different strategies are averaged on all 12 models.
>
> **Response**: Thank you for the suggestion. However, averaging over all models would be inappropriate for the purpose of Figure 3. Our benchmark studies how different retrieval strategies shape haystack construction and long-context robustness (Section 4.1), and these effects can vary substantially across models. Averaging across models could obscure whether the finding is consistent across models.
>
> **Q2**: Figure 4 is confusing. There is no comparison between retriever-ranked and random haystack ordering in the figure. It requires comparing Figure 3 and 4 to understand.
>
> **Response**: Figure 4 is not intended to present the full curves with random haystack orderings as in Figure 3. Instead, it directly visualizes the performance difference between retriever-ranked ordering and the average over three random permutations (ranked – random), allowing us to isolate the ordering effect in a single plot. The comparison is therefore internal to Figure 4, not between Figures 3 and 4. Positive values indicate a benefit from retriever-ranked ordering, while negative values indicate the opposite. We have updated the corresponding paragraph in the main text to better emphasize that Figure 4 illustrates the performance difference itself, rather than the raw curves.
>
> **Q3**: In Figure 2, the authors claim reranking with PRR consistently boosts performance, especially for multi-hop questions. However, 4-hop shows opposite results. Can you explain why?
>
> **Response**: Thank you for your question. For PPR reranking, we first use a base retriever (sparse, dense, or hybrid) to score and select the top-k documents, which then serve as seed nodes for PPR. Running PPR diffuses these initial retrieval scores over the local graph neighborhood, promoting documents that are structurally well-connected to the top-k seeds.
>
> For 2–3-hop questions, this yields consistent gains: the supporting documents are typically located close to the seed set in the graph, so PPR strengthens the structural signal and improves both recall and ranking. For 4-hop questions, the supporting documents tend to be much farther away. In this case, PPR may either not diffuse scores far enough to reach all relevant documents or introduce more false positives as the diffusion spreads over longer distances. This explains the small degradation observed for 4-hop in Figure 2. Importantly, this negative effect is marginal compared with the clear gains on 2–3-hop questions. We have updated Section 4.1 to explain this point.
>
> **Q4**: Can you show some examples on how graph-based reranking mitigates harmful distractors?
>
> **Response**: We have added an appendix section (Appendix I) with five concrete case studies demonstrating how graph-based reranking mitigates harmful distractors. This section is referenced in Section 4.1 when discussing the impact of graph-based retrieval.
>
> At a high level, base retrievers such as BM25 or Qwen3-0.6B often surface distractors with strong lexical or semantic overlap with the query. However, these distractors tend to be structurally isolated from the other top-ranked retrieval results. Reranking with Personalized PageRank (PPR) downweights these poorly connected distractors, eliminating them from the top-ranked retrieval results.

---

> > ### Author Response · Authors · 2025-11-27
> >
> > **Q5**: The paper complicated its simple idea with many unnecessary explanations. For example, the idea of using retrieved documents as haystacks is already well-used by most of the works, but this paper used section 3.1, 3.2, 3.4 to explain this. On the other hand, the paper should discuss novel dynamic NIAH in detail. The paper should be well written with concise data/task introduction and detailed analysis.
> >
> > **Response**: We appreciate the concern about exposition and page usage, but we respectfully disagree with the premise that our construction of retrieval-dependent haystacks is already “well-used by most of the works”. Most existing NIAH benchmarks (e.g., NIAH [1], LV-Eval [2], RULER [3], BABILong [4]) construct retrieval-independent distractors. To our knowledge, HELMET [5] is the only prior work that employs and emphasizes retrieval-based haystack construction. However, it does not provide a general mathematical formulation connecting retrieval strategy, haystack composition, and haystack ordering.
> >
> > Section 3.1 is therefore not a restatement of standard RAG, but a necessary formalization that surfaces the key design dimensions—retrieval strategy and haystack ordering. Sections 3.2 and 3.4 then motivate and justify our concrete implementations: heterogeneous retrieval strategies (sparse, dense, hybrid, graph-based), retrieval-ranked vs random haystack orderings, a large-scale networked corpus, and question answering samples. These components collectively define the evaluation setting and are essential for interpreting the empirical results.
> >
> > Our dynamic NIAH formulation (Section 3.3) and the analyses (Section 4.2) rely directly on these foundations. We believe the current structure strikes a reasonable balance between clarity, novelty, and empirical insight. It establishes the conceptual foundations of haystack engineering before presenting the static and dynamic evaluations built on top of them.
> >
> > [1] Gregory Kamradt. Needle In A Haystack - Pressure Testing LLMs.
> >
> > [2] Yuan et al. LV-Eval: A Balanced Long-Context Benchmark with 5 Length Levels Up to 256K.
> >
> > [3] Hsieh et al. RULER: What’s the Real Context Size of Your Long-Context Language Models?
> >
> > [4] Kuratov et al. BABILong: Testing the Limits of LLMs with Long Context Reasoning-in-a-Haystack.
> >
> > [5] Yen et al. HELMET: How to Evaluate Long-context Models Effectively and Thoroughly.

---

### Official Review · Reviewer_SV16 · 2025-10-30

**Soundness:** 3
**Presentation:** 2
**Contribution:** 3
**Rating:** 6
**Confidence:** 3

**Summary:**

The paper first identifies the gap between synthetic and real-world long-context evaluation settings. To address the Sim2Real gap, the authors propose a new benchmark mimicing real-world long-context reasoning tasks, including RAG and multi-round agentic reasoning.

The authors study heterogeneous retrieval and ranking methods, including sparse (BM25), dense (embedding-based), hybrid (sparse+dense), and graph-based (e.g., PPR) approaches. The benchmark includes (1) a document corpus, constructed from the entire English Wikipedia hyperlink network and (2) a QA dataset, extracted from single-hop (NQ) and multi-hop (MuSiQue) benchmarks.

The paper reports that:

1. The models' performance is degraded with longer context windows. Dense retrieval methods introduce "harder" distractor documents under larger context sizes and graph-based reranking improves downstream LLM performance. The authors also observe that ordering of the documents matters (not conclusive in which direction).

2. Under multi-round agentic settings, the models are sensitive to (1) cascading amplified errors from their own reasoning traces, (2) deviations from the original query's intent. An interesting insight is that the models are more robust to larger context windows (width) compared to more iterations (depth).

**Strengths:**

The paper identifies a critical gap between synthetic and real-world NIAH tasks. The **problem setup** is the main strength and the key contribution of the paper. Additionally,

1. The paper considers two tasks inspired from real-world long-context use cases: RAG and multi-round agentic reasoning (e.g., deep research).

2. The paper thoroughly studies the effect of different retrieval strategies and context engineering design choices (e.g., ordering) on the downstream performance.

3. Given that the dataset is released, the benchmark can be valuable for long-context NIAH-style evaluation.

**Weaknesses:**

**Agentic context engineering evaluation**

1. Why doesn't the model see the original question and its all previous queries as part of the input context? This can potentially mitigate the observed failure case of deviating from the original query and is more aligned with how the current models are used in multi-round tool-calling scenarios.

2. The current deep search pipelines often utilize tool-calling capabilities of the models: the models formulate queries to a search engine / retrieval module. This setup is more realistic and can change the reported results, as the models will differentiate between the user's query and queries to the retrieval module, resolving some of the observed failure cases.

**Presentation**

The plots are hard to interpret. Instead of reporting all the models in a single plot with multiple subplots, I would recommend:

1. Having a separate plot(s) for scale ablation (how does the scale affect the results). This ablation can be done on the Qwen models.

2. Having a separate plot(s) for number of hops: select 2-3 models and report the results broken down by the number of hops.

3. Select the main models for the main paper and move the rest to a model family ablation in the appendix.

**Ordering**

Are the documents ordered in ascending / descending order based on the retriever scores? How does this affect the performance?

When comparing ordered vs random context, do the authors observe lost-in-the-middle phenomenon reported in prior work? Additional ablations on the document ordering can clear the unconclusive (model-specific) results (Figure 4).

**Data contamination**

The authors report non-zero performance with no context (i.e., models relying on their parametric knowledge). Removing the questions that can be answered without search can improve the reliability of the results: I suspect that the effect of context on parametrically answerable and non-answerable questions would be different.

**Literature Review**

Even though the authors reference related work throughout the paper, the main literature review section seems limited, specifically for long-context agentic reasoning applications (e.g., search engine contamination, deep research under noisy retrieval, failure modes) and search benchmarks (e.g., BrowseComp, SearchArena, etc.)

**Questions:**

**Hop count**

The authors report variation in retrieval efficiency across questions requiring different number of hops. Additionally, it is reported that multi-hop questions are more robust to data contamination. These observations lead to a natural questions: how does this affect the downstream LLM performance? Do the current results still hold if we only look at single-hop or 4-hop question?

**RAG Models**

Why don't the authors evaluate stronger models on the RAG task (e.g., GPT-5, Gemini-2.5-Pro)? Do the current findings hold for current SOTA models?

**Ordering**

Are the documents ordered in ascending / descending order based on the retriever scores? How does this affect the performance?

---

> ### Author Response · Authors · 2025-11-27
>
> Thank you for your positive feedback and detailed suggestions. Below, we address your questions and concerns.
>
> **Q1**: In dynamic NIAH, why not include the original question and all prior retrieval queries in the input context and explicitly distinguish them? This could reduce the observed failures and better reflect multi-round tool-calling and deep-search pipelines.
>
> **Response**: Our original agent workflow was intentionally kept minimal and inspired by ReAct [1], a widely recognized foundational LLM agent framework. This design allows us to study a core question in isolation: *when an LLM must iteratively digest noisy, yet information-sufficient, long contexts and update its own intermediate reasoning, how robust is it to compounding distractors and early-stage mistakes?* A simple workflow provides a clean starting point that avoids conflating model behavior with manual prompt engineering.
>
> Your concern is valid: a workflow that is better aligned with multi-round tool-calling and deep-search pipelines allows reflecting practical performance and understanding if cascading errors are merely an artifact of simplistic workflows. Following your guidance, we conduct an ablation study using a more advanced workflow that always includes all previous retrieval queries in the input context and explicitly highlights the original question. Full templates are available in Appendix D.
>
> We find that:
> 1. **Cascading errors persist for the strongest models.** For state-of-the-art models such as Gemini 2.5 and GPT-5, the advanced workflow does not prevent multi-round degradation. The failure modes remain qualitatively similar to those observed under the simpler workflow.
> 2. **Earlier, weaker models may benefit substantially.** Models with weaker long-context capabilities (e.g., Llama-3.1-8B and Qwen2.5-7B) do show notable gains and, in some settings, even improve with additional rounds.
> 3. **Advanced prompting is a double-edged sword.** In the variable-round dynamic NIAH setting, where models are allowed to stop early, the weaker models’ performance drops significantly under the advanced workflow. Manual inspection suggests that the prompt becomes overly complex for these models, leading to failure in relevant information identification and aggregation.
>
> A plausible explanation is that through RL-based post-training [2], the latest reasoning models can mostly retain the original question’s intent with their refined queries and finding summaries, limiting the gains from additional manual prompting and workflow design. Conversely, weaker models benefit from explicit structure only while the prompt remains within their comprehension bandwidth. Once the prompt becomes too complex, it acts as an additional distractor.
>
> In summary, our ablation study indicates that:
>
> 1. Cascading errors are not an artifact of a simple workflow. They remain persistent for the most advanced models with the advanced workflow.
> 2. More complex manual prompting can aid weaker models, but it also increases cognitive burden and can become counterproductive in flexible multi-round settings.
>
> More broadly, the ablation study highlights the value of using a simple workflow in dynamic NIAH. A minimal agent design cleanly exposes fundamental robustness gaps between state-of-the-art closed-source models and open-source models, which would be partially obscured under heavier prompt engineering. The goal is not to “solve” this specific task, but to offer a simple, informative evaluation environment with implications for many agentic settings. In real-world agentic tasks, designing an optimal workflow or sophisticated prompting strategy can be difficult. Understanding how models behave with a minimally assisted workflow provides useful signals for anticipating their robustness in more complex applications.
>
> We have incorporated these findings and a corresponding discussion into Section 4.2 of the revised PDF.
>
> [1] Yao et al. ReAct: Synergizing Reasoning and Acting in Language Models. ICLR 2023.
>
> [2] DeepSeek-AI. DeepSeek-R1: Incentivizing Reasoning Capability in LLMs via Reinforcement Learning.

---

> ### Author Response · Authors · 2025-11-27
>
> **Q2**: The plots are hard to interpret. Instead of reporting all the models in a single plot with multiple subplots, I would recommend: 1) Having a separate plot(s) for scale ablation (how does the scale affect the results). This ablation can be done on the Qwen models. 2) Select the main models for the main paper and move the rest to a model family ablation in the appendix.
>
> **Response**: Thank you for your detailed suggestions. We understand the concern about visual density and cognitive burden, but we believe the current presentation is necessary for accurately conveying the key findings.
>
> NIAH is fundamentally a *scaling evaluation*: its primary goal is to examine how long-context performance changes as the context grows. Treating scaling as an “ablation” on a single model family (e.g., Qwen) would misrepresent this core property and obscure how scaling behaviors vary across different architectures.
>
> Several central observations in our paper, such as the effect of retrieval strategies, require examining multiple model families jointly. Separating models in isolated plots would make it more difficult to assess inter-model consistency, which is crucial for interpreting the generality of the results.
>
> This presentation choice is also aligned with established practice in NIAH benchmarks, which likewise present multi-model scaling results directly in the main text [1-4].
>
> [1] Hsieh et al. RULER: What's the Real Context Size of Your Long-Context Language Models?
>
> [2] Kuratov et al. BABILong: Testing the Limits of LLMs with Long Context Reasoning-in-a-Haystack.
>
> [3] Yuan et al. LV-Eval: A Balanced Long-Context Benchmark with 5 Length Levels Up to 256K.
>
> [4] Yen et al. HELMET: How to Evaluate Long-Context Language Models Effectively and Thoroughly.
>
> **Q3**: Why don't the authors evaluate stronger models on the RAG task (e.g., GPT-5, Gemini-2.5-Pro)? Do the current findings hold for current SOTA models?
>
> **Response**: Thank you for the question. Evaluating strong proprietary models across all retrieval strategies, haystack orderings, and context sizes would be prohibitively expensive. For reference, GPT-5 costs 1.25 dollars per 1M input tokens. Running the full static NIAH evaluation—6 retrieval strategies (BM25, dense, hybrid + PPR variants), 4 document orderings (retriever-ranked + 3 random permutations), 500 questions, and 5 context sizes (8K–128K)—already amounts to roughly 4,000 dollars per model, not including output-token costs. Given this constraint, we prioritized using the strongest models in the dynamic NIAH setting, where their reasoning ability and multi-round robustness are most relevant.
>
> To directly address your question, we additionally evaluate GPT-5 on static NIAH using a representative configuration (hybrid retrieval with retriever-ranked haystack ordering), with and without graph-based reranking:
>
> |                      | No Distraction | 128K |
> | -------------- | ---------------- | ------ |
> | with PPR      | 77.28              | 73.64 |
> | without PPR | 77.28              | 73.76 |
>
> GPT-5 shows minimal degradation up to 128K tokens, and PPR yields negligible downstream differences due to task saturation. This near-perfect static performance reinforces our motivation for dynamic NIAH: for SOTA models, static NIAH is largely solved, and the real remaining challenges emerge in dynamic, agentic NIAH.
>
> **Q4**: The authors report variation in retrieval efficiency across questions requiring a different number of hops. Additionally, it is reported that multi-hop questions are more robust to data contamination. These observations lead to natural questions: how does this affect the downstream LLM performance? Do the current results still hold if we only look at single-hop or 4-hop questions?
>
> **Response**: We added Figure 8 in the Appendix to break down static NIAH performance by question hop count for Llama-3.1-8B-Instruct and Qwen2.5-7B-Instruct-1M. Although the hop-specific subsets exhibit greater variance due to their smaller size, the qualitative pattern remains mostly consistent across hop counts at large context sizes: (1) dense retriever tends to introduce more challenging distractors than sparse retriever; (2) graph-based reranking provides gains, indicating a persistent benefit in mitigating harmful distractors.

---

> > ### Author Response · Authors · 2025-11-27
> >
> > **Q5**: Are the documents ordered in ascending / descending order based on the retriever scores? How does this affect the performance? When comparing ordered vs random context, do the authors observe lost-in-the-middle phenomenon reported in prior work? Additional ablations on the document ordering can clear the inconclusive (model-specific) results (Figure 4).
> >
> > **Response**: The documents are ordered in retriever-ranked descending order, following standard RAG practice.
> >
> > The ``lost-in-the-middle’’ phenomenon refers to the significant degradation in LLM performance when relevant information appears in the middle of a long context [1]. To examine its potential connection to our previous observations regarding the impact of haystack orderings, we conduct an additional ablation study using two representative models, Gemma-3-12B-IT and Gemini 2.5 Flash, on 128K-token contexts constructed with the hybrid retriever.
> >
> > We evaluate four haystack orderings:
> > - Retriever-ranked descending
> > - Retriever-ranked ascending
> > - Average over three random permutations
> > - Middle, where the ground-truth supporting documents are manually placed at the center of the context.
> >
> > |                               | descending | ascending | random | middle |
> > | -------------------- | ------------- | ------------ | --------- | -------- |
> > | Gemma-3-12B-IT | **44.1**       | 39.97        | 32.51     | 29.36   |
> > | Gemini 2.5 Flash  | 63.28           | **65.44**  | 64.06     | 65.31   |
> >
> > Gemma-3-12B-IT shows substantial sensitivity to ordering. Both descending and ascending rankings outperform the random and middle placements, and the middle condition yields the worst results, consistent with the “lost in the middle” effect. In contrast, Gemini 2.5 Flash remains robust across all orderings with minimal variation.
> >
> > These results clarify the model-specific pattern noted in Figure 4. Models that benefit most from retriever-ranked descending ordering are also the ones that suffer most from the ``lost-in-the-middle’’ issue.
> >
> > We have updated Section 4.1 to explicitly note our use of descending ordering and to include this ablation study.
> >
> > [1] Liu et al. Lost in the Middle: How Language Models Use Long Contexts.
> >
> > **Q6**: The authors report non-zero performance with no context (i.e., models relying on their parametric knowledge). Removing the questions that can be answered without search can improve the reliability of the results: I suspect that the effect of context on parametrically answerable and non-answerable questions would be different.
> >
> > **Response**: We agree that parametric knowledge and potential data contamination can influence the benchmark results. However, filtering questions based on whether they are “parametrically answerable” is difficult to implement in a fair and model-agnostic way. Different models can answer different subsets of questions without context; removing all such questions across models would leave too few examples, while filtering a different subset for each model would undermine apples-to-apples comparisons.
> >
> > Furthermore, even when a question is not fully solvable from parametric knowledge alone, models often retain partial knowledge about intermediate entities or sub-questions. For example, in “What continent is the country encompassing Luahoko located in?”, a model may fail to produce the correct final answer (“Oceania”) yet still know that Luahoko is an island, that it is encompassed by Tonga, and what the possible continents are. This makes it difficult to draw a clean boundary between parametrically answerable and non-answerable questions.
> >
> > For these reasons, we retain the full question set and instead report no-context performance as a reference.
> >
> > **Q7**: Even though the authors reference related work throughout the paper, the main literature review section seems limited, specifically for long-context agentic reasoning applications (e.g., search engine contamination, deep research under noisy retrieval, failure modes) and search benchmarks (e.g., BrowseComp, SearchArena, etc.)
> >
> > **Response**: Thank you for the suggestion. We’ve revised the related work section (Section 2) to discuss the suggested benchmarks and issues for agentic search.

---

### Official Review · Reviewer_Q1KQ · 2025-11-01

**Soundness:** 3
**Presentation:** 3
**Contribution:** 3
**Rating:** 6
**Confidence:** 2

**Summary:**

This paper introduces a new benchmark HaystackCraft built on Wikipedia hyperlink network to evaluate how heterogeneous retrieval strategies affect distractor composition, haystack ordering and LLM performance. Evaluation results using 15 long-context models highlight robust agentic long-context reasoning is far from solved.

**Strengths:**

+ a new benchmark designed to evaluate how heterogeneous retrieval strategies affect distractor composition, haystack ordering and LLM performance
+ interesting findings from the evaluation using 15 long-context models

**Weaknesses:**

- lack of deep analysis on the underlying reasons of the performance of the models over the HaystackCraft benchmark

**Questions:**

It is interesting that the current LLMs are more robust to noisy long contexts than noisy reasoning iterations. What are the possible reasons for such an observation? How can it be leveraged to improve the performance of LLMs?

---

> ### Author Response · Authors · 2025-11-27
>
> Thank you for your positive feedback. Below, we address your questions and concerns.
>
> **Q1**: It is interesting that the current LLMs are more robust to noisy long contexts than noisy reasoning iterations. What are the possible reasons for such an observation? How can it be leveraged to improve the performance of LLMs?
>
> **Response**: A plausible explanation is that current LLMs have been trained far more extensively on single-round noisy long-context settings, motivated by existing long-context benchmarks such as static NIAH. Consequently, models have learned to better tolerate noisy long inputs but remain more fragile to self-generated distractors across iterations, where early errors introduced by noisy contexts can compound and propagate during reasoning.
>
> This suggests that improving LLM robustness requires 1) developing more benchmarks that explicitly evaluate noisy multi-round reasoning and training protocols that expose models to such iterative settings. HaystackCraft is the first benchmark to reveal this width–depth robustness gap and provides a concrete testbed for developing more reliable agentic LLMs.
>
> **Q2**: Can you perform deeper analyses on the underlying reasons of the model performance over the HaystackCraft benchmark?
>
> **Response**: In the revision, we added several analyses that directly target the mechanisms behind the observed behavior in both the static and dynamic settings.
>
> **Impact of graph-based reranking in mitigating harmful distractors**. At a high level, base retrievers such as BM25 or Qwen3-0.6B often surface distractors with strong lexical or semantic overlap with the query. However, these distractors tend to be structurally isolated from the other top-ranked retrieval results. Reranking with Personalized PageRank (PPR) downweights these poorly connected distractors, eliminating them from the top-ranked results.
>
> Appendix I presents five concrete case studies illustrating how graph-based reranking mitigates harmful distractors. We also added Figure 8 in the Appendix to break down static NIAH performance by question hop count for Llama-3.1-8B-Instruct and Qwen2.5-7B-Instruct-1M. The breakdown shows that the benefit of graph-based reranking in mitigating harmful distractors remains largely consistent across hop counts.
>
> **Haystack ordering and lost in the middle**. The “lost-in-the-middle” phenomenon refers to the significant degradation in LLM performance when relevant information appears in the middle of a long context [1]. To examine its potential connection to our observations on haystack orderings, we conduct an ablation study using two representative models, Gemma-3-12B-IT and Gemini 2.5 Flash, on 128K-token contexts constructed with the hybrid retriever.
>
> We evaluate three haystack orderings:
> - Retriever-ranked descending
> - Average over three random permutations
> - Middle, where the ground-truth supporting documents are manually placed at the center of the context.
>
> |                              | descending | random | middle    |
> | -------------------- | ------------- | --------- | ---------- |
> | Gemma-3-12B-IT | **44.1**      | 32.51     | 29.36      |
> | Gemini 2.5 Flash  | 63.28         | 64.06     | **65.31** |
>
> Gemma-3-12B-IT shows substantial sensitivity to ordering. The descending ordering significantly outperforms the rest ones, and the middle condition yields the worst results, consistent with the “lost in the middle” effect. In contrast, Gemini 2.5 Flash remains robust across all orderings with minimal variation. These results clarify the model-specific patterns in Figure 4. Models that benefit most from retriever-ranked descending ordering are also the ones that suffer most from the “lost-in-the-middle’’ issue.
>
> [1] Liu et al. Lost in the Middle: How Language Models Use Long Contexts.
>
> **Sensitivity of dynamic NIAH results to workflow design**. We further run an ablation with a more advanced agent workflow that (i) explicitly highlights the original question, (ii) includes the full history of queries and analyses, and (iii) prompts explicit reflection and correction. See Appendix D for the detailed prompts.
>
> Cascading errors persist for the strongest models (Gemini 2.5, GPT‑5), indicating that the failures are not an artifact of a minimal agent design. Weaker models can benefit from the added structure, but only up to the point where the prompt itself does not become an additional source of distraction. In the variable-round setting, the extra complexity can hurt their ability to identify and aggregate relevant information. See Section 4.2 for the details.
>
> Taken together, these analyses provide a deeper explanation of HaystackCraft performance: they show how retrieval biases and graph structure shape the difficulty of the haystack, how model-specific positional biases lead to different sensitivities to ordering, and how agent workflow design interacts with models’ self-correction abilities in dynamic NIAH.

---

### Official Review · Reviewer_7Rcq · 2025-11-03

**Soundness:** 2
**Presentation:** 3
**Contribution:** 2
**Rating:** 4
**Confidence:** 4

**Summary:**

The paper argues that standard "Needle-in-a-Haystack" (NIAH) benchmarks are insufficient for evaluating long-context LLMs. The authors claim these synthetic tests fail to model two critical, real-world sources of noise: (1) biased distractors from heterogeneous retrieval systems (like in RAG) and (2) cascading, self-generated errors from multi-step agentic workflows.

To address this, they propose "Haystack Engineering," a principle for constructing more realistic noisy contexts. They instantiate this with a new benchmark, "HaystackCraft," built on the Wikipedia hyperlink network and multi-hop questions. The benchmark includes both "static" tests (evaluating different retrievers like sparse, dense, and graph-based) and "dynamic" tests (evaluating multi-round agentic reasoning).

The paper's main finding is that even SOTA models (e.g., GPT-5, Gemini 2.5 Pro) are brittle in these dynamic, agentic settings, suffering from cascading errors and an inability to self-correct.

**Strengths:**

1. The critique of synthetic NIAH tests is spot-on. We need evaluations that look more like real-world RAG and agent systems.

2. The dynamic test cleverly simulates how an agent can create its own errors.

3. Comparing sparse, dense, and graph-based retrievers is practical and shows how much the "haystack" composition matters.

4. The results, especially showing that graph-based reranking (PPR) helps a lot, are genuinely useful for practitioners.

**Weaknesses:**

1. Is this "Agent" too simple? The "agent" here is just a simple "summarize + refine query" loop. It's not clear if the findings about "cascading errors" apply to all agentic reasoning, or just to this one very simple design. A more robust agent architecture might not fail this way.

2. The paper concludes "width is safer than depth" (more context > more steps). But is the model failing because of the steps (depth), or because the underlying multi-hop question is just too hard? It's hard to tell if this is a failure of agentic reasoning or just a failure at a complex task.

3. The benchmark is only based on Wikipedia. Real-world systems run on messy, domain-specific data (like legal, medical, or internal wikis). It's a big question mark whether these findings generalize beyond a clean, well-structured corpus.

**Questions:**

1. How sensitive are these "cascading errors" to your specific prompt? Could a better-worded prompt that, for example, tells the model to "doubt its previous findings" or "consider alternatives" prevent these failures?

2. What happens if you run this dynamic test on a simple single-hop task? If the models still fail, you've proven "depth" is the problem. If they succeed, it suggests the failure is just a mix of "depth" and "task complexity."

3. Why do you believe models failed to stop early? Is this a failure of in-context learning (they didn't understand the "stop" instruction), or is it a more fundamental failure of the model's self-assessment (i.e., it thought it needed more info, even when it didn't)? Could this be "fixed" with a different prompt or a small number of few-shot examples?

---

> ### Author Response · Authors · 2025-11-27
>
> Thank you for acknowledging the strengths of our work and for the constructive feedback. Below, we address your questions and concerns.
>
> **Q1**: For the dynamic NIAH test, why did you adopt only a simple “summarize + refine query” agent design? Are the observed cascading errors specific to this minimal workflow? Would a more robust agent with stronger reflective prompting (e.g., “doubt its previous findings” or “consider alternatives”) avoid these failures?
>
> **Response**: Our original agent workflow was intentionally kept minimal and inspired by ReAct [1], a widely recognized foundational LLM agent framework. This design allows us to study a core question in isolation: *when an LLM must iteratively digest noisy, yet information-sufficient, long contexts and update its own intermediate reasoning, how robust is it to compounding distractors and early-stage mistakes?* A simple workflow provides a clean starting point that avoids conflating model behavior with manual prompt engineering.
>
> Your concern is valid: if advanced prompting is sufficient to avoid the cascading errors, the value of our contribution would be limited. Following your guidance, we conduct an ablation study using a more advanced workflow that employs **explicit reflective prompting**. The prompt includes the instruction “Based on the latest information, reflect on your earlier analyses. Update or correct them as needed.” Full templates are available in Appendix D.
>
> We find that:
>
> 1. **Cascading errors persist for the strongest models.** For state-of-the-art models such as Gemini 2.5 and GPT-5, advanced reflective prompting does not prevent multi-round degradation. The failure modes remain qualitatively similar to those observed under the original simpler workflow.
> 2. **Earlier, weaker models may benefit substantially.** Models with weaker long-context capabilities (e.g., Llama-3.1-8B and Qwen2.5-7B) do show notable gains under reflective prompting and, in some settings, even improve with additional rounds.
> 3. **Advanced prompting is a double-edged sword.** In the variable-round dynamic NIAH setting, where models are allowed to stop early, the weaker models’ performance drops significantly under the advanced workflow. Manual inspection suggests that the prompt becomes overly complex for these models, leading to failure in relevant information identification and aggregation.
>
> A plausible explanation is that the latest reasoning models have already acquired reflective behaviors through RL-based post-training [2], limiting the gains from additional manual prompting. Conversely, weaker models benefit from explicit structure only while the prompt remains within their comprehension bandwidth. Once the prompt becomes too complex, it acts as an additional distractor.
>
> In summary, our ablation study indicates that:
>
> 1. Cascading errors are not an artifact of a simple workflow. They remain persistent for the most advanced models under reflective prompting.
> 2. More complex manual prompting can aid weaker models, but it also increases cognitive burden and can become counterproductive in flexible multi-round settings.
>
> More broadly, the ablation study highlights the value of using a simple workflow in dynamic NIAH. A minimal agent design cleanly exposes fundamental robustness gaps between state-of-the-art closed-source models and open-source models, which would be partially obscured under heavier prompt engineering. The goal is not to “solve” this specific task, but to offer a simple, informative evaluation environment with implications for many agentic settings. In real-world agentic tasks, designing an optimal workflow or sophisticated prompting strategy can be difficult. Understanding how models behave with a minimally assisted workflow provides useful signals for anticipating their robustness in more complex applications.
>
> We have incorporated these findings and a corresponding discussion into Section 4.2 of the revised PDF.
>
> [1] Yao et al. ReAct: Synergizing Reasoning and Acting in Language Models. ICLR 2023.
>
> [2] DeepSeek-AI. DeepSeek-R1: Incentivizing Reasoning Capability in LLMs via Reinforcement Learning.

---

> > ### Author Response · Authors · 2025-11-27
> >
> > **Q2**: From dynamic NIAH experiment results, the paper concludes "width is safer than depth" (more context > more steps). But is the model failing because of the steps (depth), or because the underlying multi-hop question is just too hard? What happens if you run this dynamic test on a simple single-hop task? If the models still fail, you've proven "depth" is the problem. If they succeed, it suggests the failure is just a mix of "depth" and "task complexity."
> >
> > **Response**: Thank you for the thoughtful question. To separate the effect of depth (more steps) from the inherent difficulty of multi-hop questions, we additionally analyze the dynamic NIAH results on the single-hop subset alone.
> >
> > As shown in Appendix Q of the revised manuscript, the same qualitative pattern persists: enforcing more reasoning rounds consistently degrades performance across models, even though the underlying questions are substantially simpler. In contrast, increasing context size alone is generally less harmful than increasing the number of rounds. Strong models such as GPT-5 and Gemini 2.5 Pro also fail to turn additional rounds into gains and often underperform their single-round baselines. These observations suggest that “depth” itself is a primary source of failure. We have updated the “Implications” paragraph in Section 4.2 to reference this analysis.
> >
> > **Q3**: For variable-round dynamic NIAH experiment results, why do you believe models failed to stop early? Is this a failure of in-context learning (they didn't understand the "stop" instruction), or is it a more fundamental failure of the model's self-assessment (i.e., it thought it needed more info, even when it didn't)?
> >
> > **Response**: Thank you for the question. Our evidence suggests that the issue is not instruction following but self-assessment. As shown in Tables 20 and 21, all models perform an early stop in a meaningful fraction of cases (≥10%), indicating that they do understand the instruction. We have updated the “Variable-Round: Self-Correction Is Difficult” paragraph in Section 4.2 to reflect this clarification.

---

### Meta-Review · Area_Chair_YczV · 2026-01-07

**Summary:**

The reviewers generally agree that the paper is well motivated and addresses an important limitation of existing synthetic needle-in-a-haystack (NIAH) benchmarks, namely their lack of realism with respect to heterogeneous retrieval and multi-round, agentic settings. The proposed benchmark surfaces several interesting empirical observations, including how different retrieval strategies shape distractor composition and how model performance degrades under multi-round reasoning with noisy contexts. At the same time, reviewers raised concerns regarding the interpretability, generality, and clarity of the results. In particular, it remains unclear whether the observed cascading failures in the dynamic setting reflect fundamental limitations of agentic long-context reasoning or are influenced by specific workflow and prompting choices. Additional concerns included the generalization of results beyond Wikipedia, the strength of causal claims such as “width is safer than depth,” the depth of mechanistic analysis, and presentation clarity.

Iterative retrieval and RAG-style methods are widely regarded as core components of agentic context engineering. The benchmark, however, covers only evaluations of narrow range of such methods, limiting its ability to support broad conclusions about agentic context robustness. Both the static and dynamic evaluations focus on a small set of retrieval strategies and relatively simple multi-round querying workflows, while abstracting away many established approaches in iterative retrieval and RAG. As a result, it is difficult to determine whether the reported behaviors are representative of agentic context construction more broadly or are largely driven by the particular design choices explored in this benchmark.

Overall, while the paper has a strong motivation and presents some useful empirical findings, the current scope of the benchmark does not yet provide a sufficiently comprehensive or representative evaluation framework for drawing general conclusions about agentic context engineering. Broadening the range of evaluated methods and strengthening the connection between benchmark design and the paper’s higher-level claims would be important steps toward improving the work.

**Reviewer Concerns:**

The rebuttal addresses several reviewer concerns through additional analyses and clarifications, particularly around the interpretation of results in the dynamic NIAH setting. The authors provide ablations with more complex agent workflows and reflective prompting, and report that performance degradation across multiple rounds persists even for stronger models. They also include analyses intended to separate the effect of reasoning depth from task difficulty, discuss early stopping behavior, and add further breakdowns on retrieval strategies, hop counts, and haystack ordering. These additions help clarify the authors’ interpretation of the results and improve the internal consistency of the evaluation, although some conclusions remain tied to specific design choices and experimental settings

At the same time, several concerns remain insufficiently resolved. The benchmark and conclusions continue to rely on a Wikipedia-only corpus, leaving generalization to messier, domain specific settings untested. The dynamic evaluation still abstracts away from more realistic tool calling or modular agent pipelines used in practice, and the added workflows do not fully close this gap. Presentation issues raised by multiple reviewers, especially figure clarity and visual complexity, were largely defended rather than substantially revised. Overall, while the rebuttal clarifies the authors’ position, it does not fully resolve concerns about external validity, realism, and presentation quality, which limits the strength of the contribution.

**Reviewer Scores:**

Reviewer 7Rcq: partially addressed the concerns, maintained or slightly increased the score

Reviewer Q1KQ: partially addressed the concerns, maintained the positive score

Reviewer SV16: partially addressed the concerns, maintained or slightly increased the score

Reviewer L4UT: partially addressed the concerns, maintained the score

---

### Decision · Program_Chairs · 2026-01-26

Reject